# Elevated PDK1 Expression Drives PI3K/AKT/MTOR Signaling Promotes Radiation-Resistant and Dedifferentiated Phenotype of Hepatocellular Carcinoma

**DOI:** 10.3390/cells9030746

**Published:** 2020-03-18

**Authors:** Oluwaseun Adebayo Bamodu, Hang-Lung Chang, Jiann-Ruey Ong, Wei-Hwa Lee, Chi-Tai Yeh, Jo-Ting Tsai

**Affiliations:** 1Department of Hematology and Oncology, Cancer Center, Taipei Medical University-Shuang Ho Hospital, New Taipei City 235, Taiwan; dr_bamodu@yahoo.com (O.A.B.); ctyeh@s.tmu.edu.tw (C.-T.Y.); 2Department of Medical Research and Education, Taipei Medical University-Shuang Ho Hospital, New Taipei City 235, Taiwan; 3Department of General Surgery, En Chu Kong Hospital, New Taipei City 237, Taiwan; changhl0321@gmail.com; 4Department of Health Care Management, Yuanpei University of Medical Technology, Hsinchu 300, Taiwan; 5Department of Emergency Medicine, School of Medicine, Taipei Medical University, Taipei City 110, Taiwan; 12642@s.tmu.edu.tw; 6Department of Emergency Medicine, Taipei Medical University-Shuang Ho Hospital, New Taipei City 235, Taiwan; 7Department of Pathology, Taipei Medical University-Shuang Ho Hospital, New Taipei City 235, Taiwan; whlpath97616@s.tmu.edu.tw; 8Department of Medical Laboratory Science and Biotechnology, Yuanpei University of Medical Technology, Hsinchu City 300, Taiwan; 9Department of Radiology, School of Medicine, College of Medicine, Taipei Medical University, Taipei City 110, Taiwan; 10Department of Radiology, Taipei Medical University-Shuang Ho Hospital, New Taipei City 235, Taiwan; 11Graduate Institute of Clinical Medicine, College of Medicine, Taipei Medical University, Taipei City 110, Taiwan

**Keywords:** hepatocellular cancer, HCC, LIHC, PDK1, PI3K/AKT/mTOR pathway, BX795, selective inhibitor, radiotherapy, radioresistance, combination therapy, stemness, DNA damage

## Abstract

Resistance to radiotherapy (IR), with consequent disease recurrence, continues to limit the efficacy of contemporary anticancer treatment for patients with hepatocellular carcinoma (HCC), especially in late stage. Despite accruing evidence implicating the PI3K/AKT signaling pathway in cancer-promoting hypoxia, cancerous cell proliferation and radiotherapy-resistance, it remains unclear which molecular constituent of the pathway facilitates adaptation of aggressive HCC cells to tumoral stress signals and drives their evasion of repeated IR-toxicity. This present study investigated the role of PDK1 signaling in IR-resistance, enhanced DNA damage repair and post-IR relapse, characteristic of aggressive HCC cells, while exploring potential PDK1-targetability to improve radiosensitivity. The study employed bioinformatics analyses of gene expression profile and functional protein–protein interaction, generation of IR-resistant clones, flow cytometry-based ALDH activity and side-population (SP) characterization, siRNA-mediated loss-of-PDK1function, western-blotting, immunohistochemistry and functional assays including cell viability, migration, invasion, clonogenicity and tumorsphere formation assays. We showed that the aberrantly expressed PDK1 characterizes poorly differentiated HCC CVCL_7955, Mahlavu, SK-HEP1 and Hep3B cells, compared to the well-differentiated Huh7 or normal adult liver epithelial THLE-2 cells, and independently activates the PI3K/AKT/mTOR signaling. Molecular ablation of PDK1 function enhanced susceptibility of HCC cells to IR and was associated with deactivated PI3K/AKT/mTOR signaling. Additionally, PDK1-driven IR-resistance positively correlated with activated PI3K signaling, enhanced HCC cell motility and invasiveness, augmented EMT, upregulated stemness markers ALDH1A1, PROM1, SOX2, KLF4 and POU5F1, increased tumorsphere-formation efficiency and suppressed biomarkers of DNA damage—RAD50, MSH3, MLH3 and ERCC2. Furthermore, the acquired IR-resistant phenotype of Huh7 cells was strongly associated with significantly increased ALDH activity, SP-enrichment, and direct ALDH1-PDK1 interaction. Moreover, BX795-mediated pharmacological inhibition of PDK1 synergistically enhances the radiosensitivity of erstwhile resistant cells, increased Bax/Bcl-2 apoptotic ratio, while suppressing oncogenicity and clonogenicity. We provide preclinical evidence implicating PDK1 as an active driver of IR-resistance by activation of the PI3K/AKT/mTOR signaling, up-modulation of cancer stemness signaling and suppression of DNA damage, thus, projecting PDK1-targeting as a putative enhancer of radiosensitivity and a potential new therapeutic approach for patients with IR-resistant HCC.

## 1. Introduction

Liver cancer with 841,080 new cases and 781,631 disease-specific deaths in 2018 alone, ranks as the 6th most diagnosed malignancy, and 4th commonest cause of cancer-related mortality globally [1]. Histologically, liver cancer is subclassified as focal nodular hyperplasia (FNH), cholangiocarcinoma (CC), hepatocellular adenoma (HCA), hepatocellular carcinoma (HCC) and combined HCC-CC [2]. Hepatocellular carcinoma, with an increasing annual incidence and arising mostly (90%) in the context of chronic liver disease, such as underlying liver cirrhosis and chronic hepatitis B or C, accounts for a significant 75% of all liver cancer incidence and is associated with very poor survival rates, especially as patients present in late stage, with comorbidities, micro- and/or macrovascular invasion, multicentric or multifocal large tumors, organ shortage and are thus inoperable [2,3,4]. The post-diagnosis median survival time of patients with inoperable disease is 6–20 months, while the 5-year survival remains <5% [1].

Given the complex nature of HCC, therapeutic decisions in HCC clinics are dependent on disease staging, location and size of tumor, presence and extent of extra-hepatic spread and underlying hepatic function. Currently, the preferred curative modalities for patients with HCC are surgical resection and orthotopic liver transplantation (OLT), however, for patient not meeting the criteria for curative therapy, such as those with unresectable tumors, treatment options include systemic chemotherapy, molecularly targeted therapies, transarterial chemoembolization (TACE), percutaneous ethanol injection (PEI), cryoablation and various forms of IR, namely, microwave ablation (MWA), radiofrequency ablation (RFA), radioembolization, stereotactic body radiotherapy (SBRT) and external beam radiation therapy (EBRT) [4,5,6]; these are fraught with enhanced risk of severe drug-related adverse events (AEs), acquired resistance to anticancer therapeutics and IR, and relatively dismal survival benefits [5,6], thus, necessitating concerted screening for or development of novel highly efficacious therapeutics and/or the discovery of new actionable molecular oncotargets, which inhibit disease progression, alleviate resistance to treatment and improve prognosis in patients with pancreatic ductal adenocarcinoma (PDAC).

The role of IR in the treatment of HCC continues to evolve, especially with regards to technological advancement and in the context of combinatorial therapy, aimed at enhancing IR safety and efficacy, nevertheless, the anti-HCC efficacy of IR is non-apparent in the intrinsically IR-resistant cells or blunted over time by the acquisition of IR-resistance, subsequently resulting in disease relapse and poor prognosis [7,8]. Thus, the need for continued unraveling of the mechanistic underlining of IR-resistance in HCC, identification of reliable molecular targets and development of more effective therapies.

In the last decade, there has been increased implication of the phosphatidylinositol-3-kinase (PI3K)/protein kinase B (PKB, AKT)/mammalian target of Rapamycin (mTOR) signaling in the acquisition of an IR-resistant phenotype by cancerous cells of different histological origin [9,10,11,12]. Accruing evidence suggest a role for activated AKT in the prediction of sensitivity to anticancer chemo- or radiotherapy. Mechanistically, AKT plays a vital role in the PI3K/AKT/mTOR signaling cascade, however, while PI3K binds stably to the pleckstrin homology domain (PHD) finger of AKT without fully activating AKT, evidence abound that the activation of AKT through phosphorylation of the threonine residue (Thr)-308 in AKT activation loop by 3-phosphoinositide-dependent protein kinase-1 (PDK1/PDPK1) enhances AKT activity by over 100-fold, and when followed by phosphorylation of Serine (Ser)-473 at the C terminus, which is the AKT hydrophobic motif, it induces AKT by an additional 7–10-fold, and stabilizes AKT active conformation [13]. Consistent with these, PDK1 has been shown to be dysregulated in some malignancies, and accruing evidence suggesting that the underexplored PDK1 may serve as a therapeutic target and is a probable modulator of sensitivity to cancer therapy [14]. Thus, this present study investigated the role of PDK1 signaling in IR-resistance, enhanced DNA damage repair and post-IR relapse, characteristic of aggressive HCC cells, while exploring potential PDK1-targetability to improve radiosensitivity.

## 2. Materials and Methods

### 2.1. Ethics Approval and Consent to Participate

Clinical samples were collected from Taipei Medical University-Shuang Ho hospital (Taipei, Taiwan). All enrolled patients gave written informed consent for their tissues to be used for scientific research. The study was approved by the Institutional Review Board (IRB) of the Taipei Medical University-Shuang Ho hospital (Taipei, Taiwan), consistent with the recommendations of the declaration of Helsinki for biomedical research (Taipei Medical University-Shuang Ho hospital, Taiwan) and followed standard institutional protocol for human research. This study was approved by the Institutional Human Research Ethics Review Board (TMU-JIRB No. 201302016) of Taipei Medical University.

### 2.2. Access and Analysis of Public Cancer Datasets

The public online cancer data repositories used in this study include Oncomine, The Cancer Genome Atlas (TCGA), Gene Expression Omnibus (GEO) and Broad Institute Cancer Cell Line Encyclopedia (CCLE). We probed the Wurmbach liver (HG-U133_Plus_2) Affymetrix Human Genome U133 Plus 2.0 Array dataset (*n* = 75) using the Oncomine platform (https://www.oncomine.org/resource/main.html#v:18). We also used the Affymetrix Human Genome U133 Plus 2.0 Array dataset GSE6465/GPL570 analyzing the high-throughput gene expression profile of hepatocellular carcinoma xenografts (*n* = 53 samples, 54,675 genes), from the Gene Expression Omnibus (GEO) using the National Center for Biotechnology Information (NCBI) GEO Data Browser (https://www.ncbi.nlm.nih.gov/geo/geo2r/?acc=GSE6465 &platform= GPL570).

### 2.3. Drug and Reagents

BX-795 hydroxide (#SML0694, HPLC ≥ 98%) was purchased from Sigma Aldrich Co. (St. Louis, MO, USA). Stock solutions of 1 mM were dissolved in dimethyl sulfoxide (DMSO) at 15 mg/mL, and stored in dark room at −20 °C. Phosphate buffered saline (PBS, #P7059), dimethyl sulfoxide (DMSO, #D2650), sulforhodamine B (SRB) reagent (#230162), trypsin/ethylenediaminetetraacetic acid (Trypsin-EDTA, #T4049) solution, trisaminomethane (Tris) base (#93352) and acetic acid (#695092) were purchased from Sigma Aldrich Co. (St. Louis, MO, USA), while Gibco^TM^ Dulbecco’s modified Eagle’s medium (DMEM) was purchased from Invitrogen (#11966025, Invitrogen Life Technologies, Carlsbad, CA, USA).

### 2.4. Cell lines and Culture

The human HCC SK-HEP1 (ATCC^®^ HTB-52™) and normal adult liver epithelial THLE-2 (ATCC^®^ CRL-2706™) cells were obtained from American Type Culture Collection (ATCC. Manassas, VA, USA), Huh7 (JCRB0403) from the NIBIOHN ((National Institute of Biomedical Innovation, Health and Nutrition, Japanese Collection of Research Bioresources (JCRB) Cell Bank, Japan)), while FOCUS, Mahlavu, Hep3B cells were also purchased from ATCC. All cells were cultured in DMEM (Invitrogen Life Technologies, Carlsbad, CA, USA), supplemented with 10% fetal bovine serum (FBS, #16140071) and 1% penicillin-streptomycin (Invitrogen, Life Technologies, Carlsbad, CA, USA) in 5% humidified CO_2_ incubator at 37 °C. Cells were subcultured at full confluence or media changed every 48–72 h. The cell lines were identified and authenticated based on karyotype and short tandem repeat analyses by the vendors and were regularly checked and confirmed free from any mycoplasma contamination. The cells were subjected to treatment with indicated IR dosage and/or concentrations of BX795.

### 2.5. Immunohistochemistry (IHC) Analysis

For immunohistochemistry (IHC), tissue microarray (TMA) slides of the TMU-SHH HCC cohort were established, then heat-based antigen retrieval was performed in EDTA-containing buffer, sections blocked with 5% bovine serum albumin (BSA)/1% HISS/0.1% Tween20 solution and incubated with primary recombinant antibody against PDK1 (1:400 dilution; Anti-PDK1 antibody, ab90444) overnight, at 4 °C. PDK1 immunoreactivity/positivity was detected using the mouse IgGk light chain binding protein conjugated to horseradish peroxidase m-IgG BP-HRP (#sc-516102; Santa Cruz Biotechnology, Inc., Santa Cruz, CA, USA) and the EXPOSE mouse and rabbit specific HRP/DAB detection IHC kit (#ab80436, Abcam plc., Cambridge, MA, USA). This study was approved by the Institutional Human Research Ethics Review Board (TMU-JIRB No. 201302016) of Taipei Medical University.

### 2.6. Establishment of IR-Resistant HCC Cell Lines

In preliminary studies to determine optimal IR dose, Mahlavu, Hep3B and Huh7 cell lines were exposed to IR of 2-10 Gy for 5 consecutive days to determine the maximum tolerated dose (MTD). Based on the cell dysmorphia, cytoplasmic vacuolization, nuclei pleomorphism and cell hyperplasia in irradiated HCC cells compared to the control group, MTD was determined to be 2 Gy/per day for the five consecutive days in all 3 HCC cell lines. Thus, to establish IR-resistant cell lines, the cells were subsequently exposed to 2 Gy at 130 KV, 5.0 mA, every 48 h for 30 cycles (i.e., 60 Gy cumulative dose in 2 months), using the Faxitron^®^ CellRad X-ray cell irradiator (Precision X-ray Irradiation, North Branford, CT, USA). The viable HCC cells after the 30 IR cycles were designated IR-resistant—Mahlavu-R, Hep3B-R and Huh7-R. Culture media was changed every 48–72 h or cells subcultured if confluent. To confirm IR-resistance, the Mahlavu-R, Hep3B-R and Huh7-R alongside their control counterparts were exposed to 0.5–2 Gy single-doses of IR, then evaluated using functional assays, including cell viability and clonogenic-survival assay. The HCC-R cell survival fractions, clonogenicity and tumorsphere-formation efficacy were markedly higher compared to the HCC control cells.

### 2.7. Western Blot Analysis

After separation of 20 μg protein samples using 10% sodium dodecyl sulfate-polyacrylamide gel electrophoresis (SDS-PAGE) gel, the protein blots were transferred onto polyvinylidene fluoride (PVDF) membranes in the Bio-Rad Mini-Protein electro-transfer system (Bio-Rad Laboratories, Inc., CA, USA), followed by membrane-blocking in 5% skimmed milk in Tris-buffered saline with Tween 20 (TBST) for 1 h. Thereafter, membranes were incubated overnight at 4 °C with primary antibodies against p-PI3 Kinase p85 (Tyr458)/p55 (Tyr199; #4228S; 1:1000, Cell Signaling Technology Inc., Danvers, MA, USA), PI3 Kinase p110α (#4292S; 1:1000, Cell Signaling Technology), p-PDK1 (Ser241; #3061L; 1:1000, Cell Signaling Technology), PDK1 (#3062S; 1:1000, Cell Signaling Technology), p-AKT (Ser473; #9271L; 1:1000, Cell Signaling Technology), AKT (#4691L; 1:1000, Cell Signaling Technology), p-mTOR (Ser2448; #2971L; 1:1000, Cell Signaling Technology), mTOR (#2972S; 1: 1000, Cell Signaling Technology), E-cadherin (#3195S; 1:1000, Cell Signaling Technology), N-cadherin (#13116S; 1:1000, Cell Signaling Technology), Vimentin (#5741S; 1:1000, Cell Signaling Technology), Snail (#3879S; 1:1000, Cell Signaling Technology), Bcl-2 (#15071S; 1:1000, Cell Signaling Technology), Bax (#5023S 1:1000, Cell Signaling Technology) and β-actin (sc-69879; 1:500, Santa Cruz Biotechnology, Santa Cruz, CA, USA) in Appendix A. This was followed by incubation of membranes in appropriate secondary antibodies conjugated with horseradish peroxidise (HRP) at room temperature for 1 h, washed carefully with PBS thrice, and then protein band detection performed using the enhanced chemiluminescence (ECL) detection system (Thermo Fisher Scientific Inc., Waltham, MA, USA), and band densitometry-based quantification done using the ImageJ software (https://imagej.nih.gov/ij/).

### 2.8. Sulforhodamine B Cytotoxicity Assay

Using 96-well plates, 3 × 10^3^ HCC and/or HCC-R cells were seeded per well in quadruplicates and cultivated for 24 h. The cells were thereafter exposed to 0.5–2 Gy IR and/or indicated concentrations of BX795 for 48 h, subjected to 10% trichloroacetic acid (TCA) fixation, carefully washed with double-distilled water (ddH_2_O), then stained with 0.4% 0.4:1 (*w/v*) SRB/acetic acid solution. The unbound SRB dye was removed by carefully washing the cells with 1% acetic acid thrice before air-drying the plates. Thereafter, bound SRB dye was solubilized in 10mM Tris base, and absorbance, which is strongly correlated to the number of viable stained cells over a wide range, was read at a wavelength of 570 nm in the Molecular Devices Spectramax M3 multimode microplate reader (Molecular Devices LLC., San Jose, CA, USA).

### 2.9. Tumorsphere Formation Assay

In non-adherent 6-well plates, 5 × 10^4^ Mahlavu, Mahlavu-R, Huh7 or Huh7-R cells were seeded per well (Corning Inc., Corning, NY, USA) containing DMEM supplemented with Gibco^TM^ B-27^TM^ supplement (#17504044, Invitrogen, Carlsbad, CA, USA), 20 ng/mL basic fibroblast growth factor (bFGF; #13256029, Invitrogen, Carlsbad, CA, USA) and 20 ng/mL epidermal growth factor (EGF; #PHG0311, Invitrogen, Carlsbad, CA, USA). Cells were cultured for 12 days and formed tumorspheres ≥150 μm were counted under inverted phase contrast microscopy.

### 2.10. Colony Formation Assay

Clonogenicity was assessed as previously described [15]. Briefly, 1 × 10^3^ Mahlavu, Hep3B or Huh7 cells, with or without their IR-resistant counterparts were pre-exposed to indicated treatment regimen for 24 h were seeded per well in 6-well plates and incubated in 5% humidified CO_2_ incubator at 37 °C for 15 days. The colonies formed (>50 cells/colony) were then stained with crystal violet dye, photographed and counted.

### 2.11. Immunofluorescence (IFC) Staining

In Nunc™ Lab-Tek^TM^ II 8-well chamber slides, 2 × 10^4^ Mahlavu, Mahlavu-R, Huh7 or Huh7-R cells pre-exposed to indicated IR dosage or transfected with shPDK1 were seeded (#154534, Thermo Fisher Scientific Inc., Waltham, MA, USA) for 24 h. For SOX2, or OCT4A staining, the seeded cells were fixed with Image-iT^TM^ fixative solution (4% paraformaldehyde; #FB002, Thermo Fisher Scientific Inc., Waltham, MA, USA) at room temperature for 20 min, washed with 1X PBS, permeabilized with 0.1% Triton X-100 (#28314, Thermo Fisher Scientific Inc., Waltham, MA, USA) in 0.01 M PBS (pH 7.4) for 5 min, blocked with 0.2% bovine serum albumin for 1 h, air-dried and rehydrated in 1X PBS. The cells were then incubated with rabbit monoclonal antibody against Sox2 (#3579S, Cell Signaling Technology Inc., Danvers, MA, USA) or Oct-4A (#2840S, Cell Signaling Technology Inc., Danvers, MA, USA) diluted 1:500 in 1X PBS containing 3% normal goat serum at room temperature for 2 h, washed thrice in 1X PBS for 10 min each, and then incubated with goat anti-rabbit fluorescein isothiocyanate (FITC) IgG—conjugated secondary antibody (Jackson ImmunoResearch Inc., West Grove, PA, USA) diluted 1:500 in 1X PBS at room temperature for 1 h. Thereafter, the cells were washed in 1X PBS, mounted using Vectashield^®^ antifade mounting medium with 4′,6-diamidino-2-phenylindole (DAPI; #H-1200, Vector Laboratories, Burlingame, CA, USA) for nuclear staining. Cell images were taken under a Zeiss Axiophot fluorescence microscope (Carl Zeiss Microscopy LLC, Thornwood, NY, USA).

### 2.12. Transwell Matrigel Invasion Assay

For the evaluation of cell invasion, the modified Boyden chambers consisting of Corning^®^ Transwell^®^ membrane filter inserts (8-µm pore size; Corning Costar Corp., Cambridge, MA, USA) in 24-well tissue culture plates were used. For an invasion assay, 2 × 10^5^ Mahlavu, or Mahlavu-R cells were seeded into 200 µL serum-free DMEM medium in the upper surface of membranes coated with 100 µL Matrigel (BD Biosciences, San Jose, CA, USA) and allowed to invade toward the underside of the membrane in 24-well tissue culture plates containing 500 μL complete growth media with 10% FBS for 24 h. Non-invaded cells were removed by carefully wiping the upper side of the membrane with sterile cotton buds while the invaded cells on the underside of the membrane were fixed with ice-cold methanol, and then stained with 0.5% crystal violet dye in 20% ethanol for 30 min. Thereafter, the invaded cells were counted under a light microscope in 5 randomly selected visual fields at a magnification of ×400.

### 2.13. Scratch-Wound Healing Migration Assay

To evaluate cell migration, we used well-established protocol. Briefly, Mahlavu, or Mahlavu-R cells were seeded into 6-well plates (Corning Inc., Corning, NY, USA) containing complete growth media with 10% FBS, cultured to 98–100% confluence, then the median axes of the cell monolayers were denuded with sterile yellow pipette tips. The scratch-wound healing cum cell migration was monitored over time and images captured under a light microscope with 10× objective lens at the 0 and 48 h time-points after denudation, and then images were analyzed with the NIH ImageJ software (https://imagej.nih.gov/ij/download.html).

### 2.14. Small Interfering RNA (siRNA) Transfection

For transient silencing of PDK1, the PDK1-specific siRNA ((PDPK1 siRNA (h), sc-29448)) was purchased from Santa Cruz (Santa Cruz Biotechnology, Santa Cruz, CA, USA). Lipofectamine 2000 transfection reagent (#11668019, Thermo Fisher Scientific Inc., Waltham, MA, USA) was used for the transfection of the siRNA following the manufacturer’s protocol. Total protein extracted 48 h after transfection was used for Western blot analyses.

### 2.15. Flow Cytometry-Based Side Population Analyses

For identification of the side population (SP), after washing the HCC cells in warm DMEM supplemented with 3% FBS and 10 mmol/L Gibco^TM^ 4-(2-hydroxyethyl)-1-piperazineethanesulfonic acid (HEPES) buffer (#15630106, Invitrogen-Life Technologies, Carlsbad, CA, USA), 1 × 10^6^ cells were resuspended per mL of DMEM supplemented with 3% FBS, 10 mmol/L HEPES buffer and 5 μg/mL Hoechst 33342 dye, then incubated for 1.5 h at 37 °C with gentle agitation. Hoechst dye excitation was at 350–356 nm, while fluorescence was quantified at 424/44 nm or 620 nm for Hoechst blue or Hoechst blue, respectively. Dead cells or doublets were gated out. SP phenotype sorting gates were defined by 100 μM/L reserpine, an ABC-transporter inhibitor (#R0875, Sigma-Aldrich Corp., St. Louis, MO, USA). Single cell suspension was obtained by filtering cells through a 70 μm filter, sorted into SP and non-SP cell fractions and then analyzed in the BD FACSAria II System (BD Biosciences, San Jose, CA, USA). SP cell purity was ≥98%.

### 2.16. ALDH Aldefluor Activity

The Aldefluor^TM^ kit (#01700, Stem Cell Technologies) was used for the profiling and isolation of Huh7 or Huh7-R cells with high or low ALDH activity following the manufacturer’s instruction. Briefly, after incubating the cells in Aldefluor^TM^ assay buffer containing ALDH substrate, BODIPY-aminoacetaldehyde (BAAA) for 45 min at 37 °C. ALDH^+^ cells were delineated by the ability to catalyze BAAA to its fluorescent product, BODIPY-aminoacetate (BAA), and ALDH enzymatic activity was blocked by Aldefluor^TM^ DEAB reagent (#01705), a specific ALDH inhibitor. Fluorescence-activated cell sorting (FACS) gates were defined relative to the DEAB-treated samples-based baseline fluorescence. The cells were resuspended in fresh assay buffer after incubation. ALDH^+^ and ALDH^−^ cells were sorted in the BD FACSAria II System (BD Biosciences, San Jose, CA, USA).

### 2.17. Determination of Synergism of Combinatorial Therapy

The synergistic effect of combining IR and BX795 was evaluated by adapting the Chou-Talalay algorithm of multiple drug combination. CompuSyn software (ComboSyn Inc., Paramus, NJ, USA) was used following the guideline of two therapies combination analysis. All combination dose-points falling within the right-angled ‘isobologram’ triangle, was defined as synergism, if the dose-points laid on the hypotenuse, additivity was considered, and when the dose-points fell outside the isobologram, the combination was designated as antagonistic.

### 2.18. Statistical Analysis

All results represent the mean ± SD of assays performed at least three times in triplicates. The 2-sided Student’s *t*-test was used for intergroup comparison, while 1-way ANOVA with a Tukey’s post-hoc test was used for comparisons between multiple groups. All statistical analyses were performed using the GraphPad Prism version 7.0 for Windows (GraphPad Software, Inc., La Jolla, CA, USA). *p*-value < 0.05 was considered statistically significant.

## 3. Results

### 3.1. PDK1 Is an Independent Driver of the PI3K/AKT/mTOR Signaling Pathway and Its Aberrant Expression Characterizes Poorly Differentiated Aggressive HCC Cells

Following our in-house mining of high through-put gene expression data from public repositories, to unravel the role of the PI3K/AKT/mTOR signaling axis, and more specifically PDK1 signaling in HCC, we performed in silico proteotranscriptomic analyses of selected molecular components of the pathway in question. Results of our differential expression profiling of the Wurmbach liver cohort (*n* = 75), with 45 context-relevant sample, revealed concomitant significant up-regulation of PI3K (1.22-fold, *p* = 0.030), PDK1 (2.17-fold, *p* = 8.95 × 10^−6^), mTOR (1.37-fold, *p* = 6.26 × 10^−4^) and statistically insignificant down-regulation of AKT mRNA (−1.14-fold, *p* = 0.847) in HCC samples (*n* = 35) compared to the normal liver tissues (*n* = 10; Figure 1A). Protein–protein interaction (PPI) enrichment analyses using the STRING-db (https://string-db.org) platform for PPI network prediction revealed very strong association between PI3K, PDK1, AKT, mTOR and other indicated effectors and/or mediators of the PI3K/AKT/mTOR signaling axis, as expressed by the relatively high mean local clustering coefficient of 88.3% (*p* < 1.0 × 10^−16^ (Figure 1B)). Furthermore, results of our Western blot analyses showed that compared to its non-expression in the normal adult liver epithelial THLE-2 cells, the expression of p-PI3K p85 (Tyr458)/p55 (Tyr199), PI3K, p-PDK1 (Ser 241), PDK1, p-AKT (Ser 473), AKT, p-mTOR (Ser 2448) and mTOR proteins were highly or moderately highly enhanced in the poorly differentiated SK-HEP1 and Mahlavu, and Hep3B cell lines, moderately enhanced in poorly differentiated FOCUS (CVCL_7955) cells, but mildly or not expressed in the well-differentiated Huh7 cells (Figure 1C). Moreover, coupled with the consistent high mRNA intensity and protein expression levels of PDK1 in earlier data, from IHC-based proteome analyses using the human protein atlas platform version 18.1 (https://www.proteinatlas.org/ENSG00000152256-PDK1/pathology/tissue/liver+cancer#img) we observed a significant positive correlation between PDK1 immunoreactivity/expression and disease stage/progression (Figure 1D). These data did indicate, at least in part, that PDK1 was an independent driver of the PI3K/AKT/mTOR signaling pathway and its aberrant expression characterized poorly differentiated aggressive HCC cells (Figure 1D).

### 3.2. Altered PDK1 Expression Is Sufficient for the Deactivation of the PI3K/PDK/AKT/mTOR Oncogenic Signaling and Sensitizes Aggressive HCC Cells to Radiotherapy

Having shown that enhanced PDK1 expression plays an essential role in the PI3K/AKT/mTOR signaling axis and characterizes poorly differentiated aggressive HCC cells, to gain some insight into the mechanistic underlining of PDK1 on HCC oncogenicity and therapy response, we evaluated the effect of increasing doses of IR on different HCC cells. We observed that while 24 h exposure to 0.5–2 Gy IR significantly reduced the viability of PDK1^low^ Huh7 (46.1%, *p* < 0.05 at 2 Gy) and PDK1^low/moderate^ FOCUS cells (38.4%, *p* < 0.05 at 2 Gy), its effect on the PDK1^moderate^ Hep3B cell was mild (~24% at 2 Gy), and it had no apparent or insignificant cytocidal effect on the PDK1^high^ SK-HEP1 and Mahlavu cells, respectively (Figure 2A). Furthermore, we demonstrated that aside from eliciting a 59% decline in the viability of PDK1^high^ Mahlavu cells, siRNA-mediated loss-of-PDK1 function (siPDK1) enhanced their sensitivity to 2 Gy IR by a significant 80%; similarly in synergism with siPDK1, the cytocidal effect of 2 Gy IR was increased to 97% from 42% alone in PDK1^low^ Huh7 cells, and to 93% from 26% alone in PDK1^moderate^ Hep3B cells (Figure 2B), thus, indicating a vital role for PDK1 expression and/or activity in the radiosensitivity of HCC cells. In addition, compared to the 27%, 14% or 51% loss of clonogenicity elicited by exposure to 2 Gy IR in Mahlavu, Hep3B or Huh7 cells, respectively, siPDK1 inhibited the ability of Mahlavu, Hep3B or Huh7 cells to form colonies by 91%, 96% or 99.8%, respectively (Figure 2C). Combining siPDK1 with 2 Gy IR was incompatible with clonal survival in all three cell lines (data not provided). Interestingly, the observed inhibition of cell viability and attenuation of clonogenicity by siPDK1 alone or in synergism with 2 Gy IR, were associated with down-regulated p-PDK1, PDK1, p-PI3K p85 (Tyr458)/p55 (Tyr199), PI3K, p-AKT, AKT, p-mTOR and mTOR in the PDK1^high^ Mahlavu cells (Figure 2D). These results do indicate that altered PDK1 expression via specific targeting of PDK1 was sufficient to deactivate the PI3K/AKT/mTOR oncogenic signaling in HCC cells and sensitized the aggressive cells to IR.

### 3.3. Aberrant PDK1 Expression Is Implicated in the Acquisition of Radioresistance and Evasion of DNA Damage by HCC Cells

Due to the implication of DNA damage in the death of cancerous cells exposed to IR and the therapeutic benefit of exploiting the reduced resolution of IR-induced clustered DNA damage [16] and altered DNA damage repair (DDR) genes in liver cancer [17], firstly, we probed and reanalyzed Huynh H et al.’s E-GEOD-6465, A-AFFY-44, AFFY_HG_U133_PLUS_2 data set on the array expression profiling of xenografts of HCC (*n* = 53 samples, 54,675 genes) (https://www.ebi.ac.uk/arrayexpress/experiments/E-GEOD-6465/). Generated expression-based heat-map revealed a dichotomization of our selected gene-sets, such that the PDK1, ALDH1A1, CD133/PROM1, OCT4A/POU5F1, SOX2 and KLF4 genes clustering with TP53 were up-regulated, while the DDR genes RAD50, MSH3, MLH3, ERCC2,and BLM gene cluster were down-regulated (Figure 3A). Additional analyses using the STRING-db platform (https://string-db.org) for PPI network prediction further confirmed earlier results, as we observed a very strong association between components of the PI3K/AKT signaling, namely PDK1/PDPK1, AKT1, MTOR and stemness marker complicit in IR-resistance ALDH1Al, CD133/PROM1, OCT4A/POU5F1, SOX2 and KLF4 pooled together, while DNA damage markers RAD50, MSH3, MLH3, ERCC2 and BLM pooled together (Figure 3B). The average local clustering coefficient for the clustered proteins was 0.827 and PPI enrichment *p*-value was *p* < 2.45 × 10^−10^ (Figure 3B). Due to the suggested implication of cancer stem cell (CSCs) markers in PDK1-induced IR-resistance, we further examined if and to what extent PDK1-induced IR-resistance affects the side population (SP), which is representative of the CSCs pool in vitro. Comparative analyses of the HCC wild type and PDK1-rich IR-resistant clones (HCC-R) revealed a 1.40-fold increase in the SP in Mahlavu-R compared to its wild type counterpart; similarly compared to Hep3B or Huh7 cells, the SP was increased by 2.14-fold or 7.03-fold in Hep3B-R or Huh7-R cells, respectively (Figure 3C). More so, because of the documented implication of ALDH1 in IR-resistance [18], we probed the GDC TCGA liver cancer (LIHC, *n* = 469) for probable relationship between PDK1 and ALDH, and showed a positive correlation between ALDH1A1 and PDK1 (R = 0.27, *p* = 1.6 × 10^−8^; Figure 3D). Consistent with the ‘coexpression - function similarity’ paradigm, and in conformity with conventional knowledge that when an inactive enzyme, otherwise known as an ‘apoenzyme’, (in this case, apoALDH1A1) binds with an organic or inorganic helper-molecule/cofactor, a complete and catalytically active form of the enzyme called an ‘holoenzyme’ is formed, we generated a spatiotemporal visualization of the probable interaction between PDK1 and ALDH1 using the Schrödinger’s PyMOL molecular graphics system (https://pymol.org/2/), and demonstrated that the catalytic domain of PDK1 (protein data bank, PDB: 1H1W) binds directly with human apoALDH1A1 (PDB: 4WJ9) with an interaction score of 17.5, an atomic contact energy (ACE) of -50.05 kcal/mol, and root-mean-square deviation (RMSD) of 23.52Å (Figure 3E, also see Appendix A). In parallel assays, we observed significantly enhanced ALDH1 activity in the PDK1-rich Huh7-R cells, as demonstrated by a 24.16-fold increase in ALDH activity in the former PDK1^low^ Huh7 cells when they acquire an IR-resistant phenotype (Huh7-R; Figure 3F), which is consistent with our predicted PDK1-ALDH1 interaction probability of 0.80 or 0.99 using the random forest (RF) or support-vector machine (SVM) classifier algorithm, respectively, as shown in Figure 3D. PDK1 interacts with ALDH and directly modulate the expression and/or activity of ALDH in HCC cells. Representative Western blot image and histograms of the differential expression of PDK1 and ALDH1 in adherent wild-type Mahlavu cells or their tumorsphere counterparts (Appendix A). These data indicate, at least in part, that PDK1 directly interacts with and activates ALDH1A1, and implicates the aberrant PDK1 expression in the acquisition of IR-resistance and evasion of DNA damage by HCC cells.

### 3.4. PDK1-Dependent Radioresistance Is Associated with the Enhanced Metastatic and Cancer Stem Cell-Like Phenotypes of HCC Cells

Having implicated the aberrant expression of PDK1 in the acquisition of IR-resistance by HCC cells, we sought to rule out its complicity in the enhanced oncogenicity and CSCs-like phenotypes of the HCC cells. We demonstrated that the Mahlavu-R cells were significantly more mobile than the Mahlavu cells, as indicated by a 2.15-fold (*p* < 0.001) enhanced cell migration in the Mahlavu-R compared with the Mahlavu cells (Figure 4A). Additionally, we observed that compared to the Mahlavu cells, a significant time-dependent increase was induced in the number of invaded Mahlavu-R cells (wild-type (WT) vs. R: 24 h, 1.48-fold, *p* < 0.01; 48 h, 1.47-fold, *p* < 0.001; Figure 4B). In addition, compared to their wild type counterpart, the clonogenicity in the Huh7-R (6.21-fold, *p* < 0.001), Mahlavu-R (13.18-fold, *p* < 0.01) and Hep3B-R (5.30-fold, *p* < 0.01) was significantly enhanced (Figure 4C). We also demonstrated that the observed enhanced migration, invasion and colony formation potentials in the PDK1-rich Hep3B-R, Mahlavu-R and Huh7-R cells were concomitantly associated with marked up-regulation of p-PDK1, PDK1, N-cadherin, Vimentin and Snail protein expression levels, with converse down-regulation of E-cadherin, compared to their expression in the HCC-WT cells (Figure 4D). Moreover, the Huh7-R or Mahlavu-R cells exhibited a marked increase in the tumorsphere sizes (Huh7 vs. Huh7-R: 2.01-fold, *p* < 0.001; Mahlavu vs. Mahlavu-R: 2.48-fold, *p* < 0.001), number (Huh7 vs. Huh7-R: 3.29-fold, *p* < 0.001; Mahlavu vs. Mahlavu-R: 4.72-fold, *p* < 0.001) and formation efficiency (Huh7 vs. Huh7-R: 4.14-fold, *p* < 0.001; Mahlavu vs. Mahlavu-R: 1.83-fold, *p* < 0.01; Figure 4E). For tumorsphere formation efficiency (TFE), the formula used was $TFE=\frac{Nx}{N1}-1\;$, where N_1_ is the number of formed tumorspheres ≥150 mm in primary or first generation and N_x_ is the number of formed tumorspheres ≥150 mm in the subsequent generation. Expectedly, nuclear OCT4A and SOX2 immunoreactivity were enhanced in the Mahlavu-R and Huh7-R cells compared with their wild type counterparts (Figure 4F). These results are indicative of existent association between the PDK1-dependent IR-resistance, increased epithelial-to-mesenchymal transition (EMT) and the enhanced metastatic and CSCs-like phenotypes of HCC cells.

### 3.5. Pharmacologic Targeting of PDK1 Resensitizes HCC Cells to Radiotherapy-Induced Apoptosis Signals Dose-Dependently, and Significantly Suppress Their Oncogenicity

In the light of our accruing evidence implicating PDK1 in the enhanced IR-resistance and associated CSCs-linked oncogenicity of HCC cells, we evaluated the translatability and probable clinical feasibility of our findings by investigating the likely efficacy of pharmacologically targeting PDK1 on the IR-resistant phenotype of HCC cells, using BX795, a potent small molecule ATP-competitive inhibitor of PDK1. Our results indicate that in the presence of 1.25–10 μM enhanced the cytocidal effect of 0.5–2 Gy IR on the Mahlavu-R cells, dose-dependently (Figure 5A, upper panel). Interestingly, this synergistic effect was akin to that observed in the wild type Mahlavu subjected to same combinatorial therapeutic regimen, howbeit greater in the wild type cells (Figure 5A, lower panel). Furthermore, we performed a synergy evaluation of the BX795/IR combinatorial therapy using the Chou-Talalay isobologram algorithm, and the results were corroboratory of our earlier data, as demonstrated by all twelve BX795/IR combination dose-points lying within the isobologram (Figure 5B). In addition, we demonstrated that the observed therapeutic effect was associated with enhanced Bax/Bcl-2 apoptotic index as shown by the moderate or strong expression intensity of Bax protein in Mahlavu-R cells treated with 1.25 μM BX795 alone or 1.25 μM BX795/2 Gy IR combined, respectively, compared to the null/mild Bax protein expression in the 2 Gy IR alone or untreated control group; conversely, compared to the strong expression of Bcl-2 protein in the control or 2 Gy IR alone group, Bcl-2 protein expression was mild or null in the 1.25 μM BX795 alone or 1.25 μM BX795/2 Gy IR combined group, respectively (Figure 5C), suggesting an apoptosis-dependent mechanism for the synergistic therapeutic effect. Consistent with the above, we also observed that in comparison to the control group, treatment with 2 Gy IR, 1.25 μM BX795 or 1.25 μM BX795/2 Gy IR combined elicited a 39% (*p* < 0.05), 48.3% (*p* < 0.01) or 97% (*p* < 0.001) reduction in the number of invaded wild type Mahlavu cells, respectively, and a 20% (*p* < 0.05), 27.6% (*p* < 0.05) or 71.9% (*p* < 0.001) reduction in the number of invaded Mahlavu-R cells (Figure 5D). A similar effect trend was also observed for the effect of 2 Gy IR, 1.25 μM BX795 or 1.25 μM BX795/2 Gy IR combined on the ability of Mahlavu and Mahlavu-R cells to form colonies (Figure 5E). These data do indicate that the BX795-mediated pharmacological targeting of PDK1 resensitizes HCC cells to radiotherapy-induced apoptosis signals dose-dependently, and significantly suppress their oncogenicity.

## 4. Discussion

While the last 3 decades has been characterized by the broadening of our understanding of HCC pathobiology, and advances in diagnostic and therapeutic strategies for managing patients with HCC, therapeutic success and clinical outcome remain poor, especially as the incidence of resistance to conventional anticancer therapies and cancer recurrence after an initial response to treatment, including IR, continue to rise [1,2,3,4,5,6]. Given the critical role of oncogene addiction and/or aberrant oncogene expression in the initiation, metastasis, resistance to therapy and recurrence of HCC, oncogene addiction constitutes an ‘Achilles heel’ in any successful molecular anticancer therapy, and its targetability represents a putative therapeutic strategy with high curative efficacy and strong antirelapse potential. This necessitates the discovery of novel onco-addictive molecular targets and development of new therapeutic strategies that are efficacious against HCC oncogenicity, biomolecular drivers of HCC development, dissemination and the acquisition of a therapy-resistant phenotype, and therapy failure in patients with PDAC [19,20,21,22]. Against this background, this study unravels a role for PDK1 as one such onco-addictive driver of oncogenicity and IR-resistance.

In the present study, we demonstrated that (i) PDK1 is an independent driver of the PI3K/AKT/mTOR signaling pathway and its aberrant expression characterizes poorly differentiated aggressive HCC cells, with (ii) altered PDK1 expression being sufficient for the deactivation of the PI3K/PDK/AKT/mTOR oncogenic signaling, and sensitization of aggressive HCC cells to radiotherapy. Mechanistically, we also showed that while (iii) aberrant PDK1 expression is implicated in the acquisition of IR-resistance and evasion of DNA damage by HCC cells, (iv) PDK1-dependent IR-resistance is associated with the enhanced metastatic and cancer stem cell-like phenotypes of HCC cells and that the (v) pharmacologic targeting of PDK1 resensitizes HCC cells to radiotherapy-induced apoptosis signals dose-dependently and significantly suppresses their oncogenicity. As with all our works, these findings add to the current repertoire of knowledge on HCC pathobiology, and are posited to help shape potential therapeutic decision-making for managing patients with HCC, in the context of the contemporary clinical challenge of IR-resistance, therapy failure, disease recurrence and poor prognosis amongst patients with HCC.

In the present study, for the first time, to the best of our knowledge, we demonstrated that PDK1 is an independent driver of the PI3K/AKT/mTOR signaling pathway and its aberrant expression characterizes poorly differentiated aggressive HCC cells (Figure 1). This is consistent with reports indicating that while PDK1 oncogenic activities are not dependent on PI3K/AKT signaling, the later more often than not is modulated by PDK1 expression [23], and as rightly put by Tan J et al., while PDK1 is almost always linked with the PI3K/AKT signaling pathway, evidence abound that it does also induce other efferent oncogenic signaling, such as demonstrated by its unmediated induction of Polo-like kinase 1 (PLK1) phosphorylation, with subsequent activation and nuclear accumulation of MYC, resulting in the growth and survival of cancerous cells, induction of an embryonic stem cell (ESC)-like gene-signature, which is associated with aggressive cancer traits and robust CSC-driving signaling, as well as resistance to mTOR-targeted therapy [24]. More so, the characterization of poorly differentiated aggressive HCC cells by aberration in PDK1 expression is of clinical relevance, since it is broadly understood that poorly differentiated cancerous cells exhibit greater degree of resistance to therapy, and are strongly associated with increased metastasis and poor prognosis [25], thus, highlighting a critical role for PDK1 as a putative molecular target in HCC.

Our data indicating that the aberrant expression of PDK1 characterizes poorly differentiated aggressive HCC cells is of translational relevance, considering that PDK1 facilitates the phosphorylation of its protein substrates, and phosphorylation has been shown to modulate the maintenance or repression of pluripotency, which is the ability of individual cells to differentiate into any somatic cell lineage [26,27]. Understanding that malignant cells differ from normal cells based, among other traits, on their propensity to proliferate without terminal differentiation; we posit that the observed aberration in PDK1 expression and/or activity permits and facilitates the occurrence of perpetual/unlimited proliferation of HCC lineage-committed progenitors while deterring terminal differentiation of the cancerous cells [28]. In fact, differentiation-failure and the degree of such differentiation-failure, as reflected by the undifferentiated or poorly differentiated cell status, distinguish benign from malignant tumors and dictate the degree of malignant transformations [28]. Our findings thus indicate the complicity of PDK1 in differentiation-failure oncogenic transformation, and the therapy-resistant phenotype of HCC cells, especially as cellular differentiation depends on the activity of master pluripotency transcription factors (TFs) such as MYC, OCT4, or SOX2 and cofactors like PDK1 invariably serving as important transcriptional coactivator or corepressor that use adenosine triphosphate (ATP) for chromatin remodeling, which ‘switches on’ or ‘switches off’ substrates/target genes, enhancing the proliferative index and consequently eliciting treatment failure and poor prognosis [27,28].

In addition, we demonstrated that altered PDK1 expression is enough for the deactivation of the PI3K/PDK/AKT/mTOR oncogenic signaling and sensitizes aggressive HCC cells to radiotherapy (Figure 2). This finding aligns with our evolving understanding of the critical role of PDK1 as a cancer addictive oncogene. In the light of our findings, we posited a probable dependence of disease progression in patients with HCC on expression and/or activity aberrations in oncogene PDK1, which is reminiscent of oncogene addiction, such that the therapeutic targeting of PDK1 or selective blocking of its activities was enough and sufficient to suppress cell viability, impede proliferation, inhibit colony formation and deactivate the PI3K/AKT/mTOR signaling cascade. Having said this, our finding is consistent with the recently demonstrated role of PDK1 as a crucial regulator of cancerous cell migration, invasion and dissemination, while concomitantly modulating the activation status of several oncogenic proteins, including PI3K and AKT/PKB [29], as well as evidence indicating that altered PDK1 expression is a crucial component of the oncogenic PI3K/AKT signaling in breast cancer, and that the targeting of PDK1 sensitizes cancerous cells to therapy [30].

In addition, we demonstrated that aberrant PDK1 expression is implicated in the acquisition of IR-resistance and evasion of DNA damage by HCC cells (Figure 3). This is particularly important considering the role of dysregulated DNA damage/DNA damage repair in the death of cancerous cells exposed to IR and the exploitable therapeutic benefit of reduced resolution of IR-induced clustered DNA damage [16] and altered DNA damage repair genes in liver cancer [17]. The last 3 decades has seen a surge in advocacy for and research into the exploitation of DNA damage response proteins as potential molecular targets to enhance the anticancer effect of radiotherapy, and numerous agents or molecular events that impair key response proteins continue to be combined with radiation in clinical trials [31]. Our data demonstrating an inverse correlation between PDK1 clustered with CSCs/IR-resistance markers and DNA damage repair gene cluster is of translational relevance. This is particularly so, considering the therapeutic challenge of innate or acquired resistance to genotoxic agents or events such as topoisomerase (TOP) inhibitors and ionizing radiation, which are known to induce some of the most toxic genomic lesions—double-strand DNA breaks (DSBs) [31]. Against the background that 1 Gy IR elicits 1000 single strand breaks (SSBs) and 35 DSBs per cell [31], and extrapolating from our presented data, it is probable that the aberrant expression of PDK1 facilitates replication forks, up-regulates DSB religation while suppressing the conversion of IR-induced primary DNA damage/lesions into fatal DSBs through the deregulation of non-homologous end joining (NHEJ), which occurs during late S and G2 cell cycle phases [31]. Intriguingly, we demonstrated for the first time to the best of our knowledge that PDK1 not only interacts with ALDH1A1, but that it also activates it.

Intriguingly, our demonstration of direct interaction between the catalytic domain of PDK1 and human apoALDH1A1 is the first evidence-based documentation of the role of PDK1 in the activation of ALDH1, to the best of our knowledge (Figure 3). Herein, PDK1 serving as a cofactor, covalently binds to the inactive ALDH1A1 apoenzyme (apoALDH1A1) to form a complete and catalytically active ALDH1A1 holoenzyme (holoALDH1A1). This in part, explains the observed clustering of PDK1 with CSC markers, and their association with enhanced oncogenicity, suppressed DNA damage genes RAD50, MSH3, MLH3, ERCC2, CLK2 and BLM, with resultant resistance to radiotherapy.

We also showed that PDK1-dependent IR-resistance is associated with the enhanced metastatic and CSCs-like phenotypes of HCC cells (Figure 4 and Appendix A). This is consistent with contemporary knowledge that CSCs-like cells exhibit increased resistance to chemo- and radiotherapy compared to their non-CSCs counterparts in same tumor niche [32], even as cells undergoing EMT have been shown to exhibit resistance to genotoxic stress mediated by conventional radio- and chemotherapy [33,34], thus, linking resistance with CSCs-like and EMT-phenotypes. In fact, it was recently suggested that the acquisition of EMT and CSCs phenotypes is linked with the activation of PI3K/AKT/mTOR signaling in IR-resistant prostate cancer cells [11]. Furthermore, we also demonstrated that the BX795-mediated pharmacologic targeting of PDK1 resensitizes HCC cells to IR-induced apoptosis signals dose-dependently, and significantly suppress their oncogenicity (Figure 5), which is consistent with findings showing that PDK1 signaling is critical for the growth and survival of cancerous cells, and that small-molecule inhibition of PDK1/PLK1 activity effectively targeted and impaired MYC dependency with its associated therapy resistance [24].

## 5. Conclusions

This present study as we depicted in our schematic abstract (Figure 6), demonstrated that aberrantly expressed PDK1 independently drives PI3K/AKT/mTOR oncogenic signaling, characterizes poorly differentiated cells, activates ALDH1 and is associated with the desensitization of aggressive hepatocellular carcinoma cells to radiotherapy. Thus, we provide preclinical evidence for the pleiotropic contribution of PDK1 to the IR-resistant phenotype of aggressive HCC cells by modulating oncogenic, stemness, metastatic and DNA damage repair signaling. These findings provide important mechanistic insights into HCC biology and have significant therapeutic implications.

## Figures and Tables

**Figure 1 cells-09-00746-f001:**
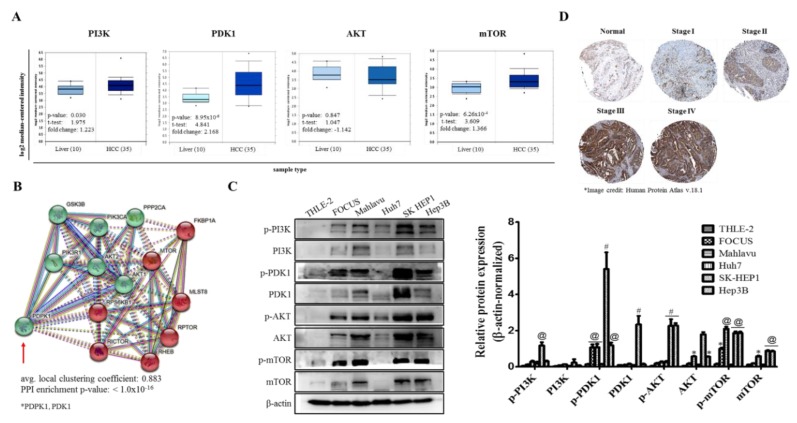
PDK1 is an independent driver of the PI3K/AKT/mTOR signaling pathway and its aberrant expression characterizes poorly differentiated aggressive hepatocellular carcinoma (HCC) cells. (**A**) Box and whisker plots of the differential expression of PI3K, PDK1, AKT and mTOR in HCC or normal liver tissues from the Wurmbach liver cohort. (**B**) STRINGdb-generated visualization of the protein–protein interaction between PDK1/PDPK1, and molecular components of the PI3K/AKT/mTOR signaling pathway. (**C**) Representative Western blot images and histograms showing the differential expression of p-PI3K p85 (Tyr458)/p55 (Tyr199), PI3K, p-PDK1 (Ser 241), PDK1, p-AKT (Ser 473), AKT, p-mTOR (Ser 2448) and mTOR proteins in THLE-2, CVCL_7955, Mahlavu, Huh7, SK-HEP1 or Hep3B cell lines. (**D**) Representative IHC images of PDK1 immunoreactivity in Stages I–IV HCC, compared to normal liver tissue. β-actin served as loading control. * *p* < 0.05, @ *p* < 0.01, # *p* < 0.001 vs. THLE-2; Green nodes, PI3K/AKT signaling; red nodes, mTOR signaling; PPI, protein–protein interaction.

**Figure 2 cells-09-00746-f002:**
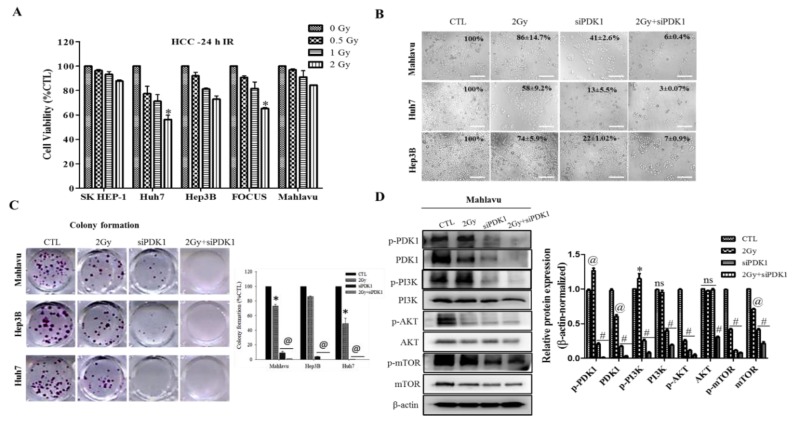
Altered PDK1 expression is enough for the deactivation of the PI3K/PDK/AKT/mTOR oncogenic signaling and sensitizes aggressive HCC cells to radiotherapy. (**A**) Histograms of the effect of 24 h exposure to 0.5–2 Gy IR on the viability of SK-HEP1, Huh7, Hep3B, CVCL_7955 or Mahlavu cell lines. (**B**) Representative photo-images of the effect of 2 Gy, siPDK1 or 2 Gy+siPDK1 on the viability/proliferation, and morphology of Mahlavu, Huh7 or Hep3B cells. Scale bar: 100 μm, 10× objective. (**C**) Representative photo-images showing the effect of 2 Gy or siPDK1 on clonogenicity of Mahlavu, Huh7 or Hep3B cells. Scale bar: 100 μm, 10× objective. (**D**) Representative western blot images and histograms showing the effect of 2 Gy, siPDK1 or 2 Gy+siPDK1 on the expression levels of p-PI3K p85 (Tyr458)/p55 (Tyr199), PI3K, p-PDK1 (Ser 241), PDK1, p-AKT (Ser 473), AKT, p-mTOR (Ser 2448) and mTOR proteins in Mahlavu cells. * *p* < 0.05, @ *p* < 0.01, # *p* < 0.001 vs. CTL; ns, not significant.

**Figure 3 cells-09-00746-f003:**
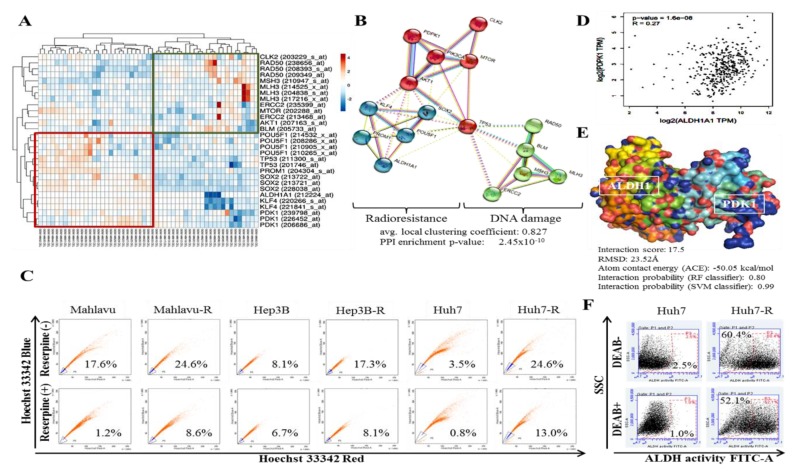
Aberrant PDK1 expression is implicated in the acquisition of radioresistance and evasion of DNA damage by HCC cells. (**A**) Heatmap of gene-set expression in the E-GEOD-6464, AFFY_HG_U133_PLUS_2 data set. Rows are centered; unit variance scaling is applied to rows. All 29 rows and 53 columns were clustered using correlation distance and average linkage. (**B**) STRINGdb-generated visualization of the protein-protein interaction between PDK1/PDPK1, PI3K/AKT/mTOR signaling pathway (red nodes), CSCs markers (blue nodes) and DNA damage markers (green nodes). (**C**) Graphical representation of the differential SP in Mahlavu, Mahlavu-R, Hep3B, Hep3B-R, Huh7 or Huh7-R cells, in the presence or absence of reserpine. (**D**) Graphical representation of the correlation between ALDH and PDK1 expression in the GDC TCGA liver cancer cohort, *n* = 469. (**E**) Schrödinger’s PyMOL molecular graphics system-generated molecular docking of PDK1 and ALDH1. (**F**) Representative images showing the differential ALDH activity in Huh7 or Huh7-R cells in the presence or absence of DEAB. DEAB, Aldefluor inhibitor; DEAB+, negative control; Reserpine served as inhibitor of Hoechst 33342 dye efflux; PPI, protein–protein interaction; RSMD, root-mean-square deviation; TPM, transcript per million.

**Figure 4 cells-09-00746-f004:**
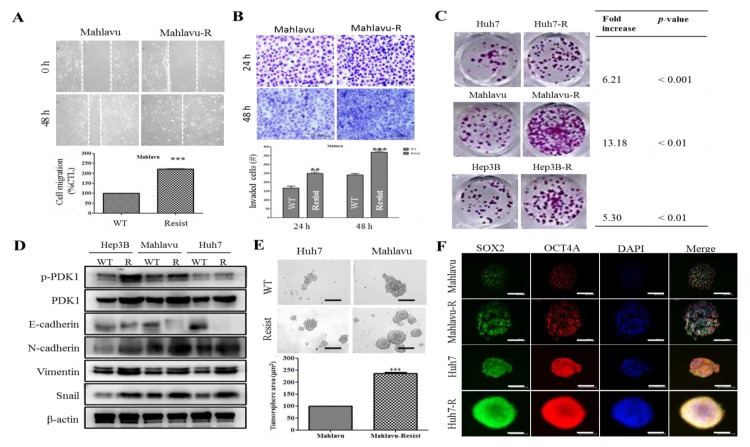
PDK1-dependent radioresistance is associated with the enhanced metastatic and cancer stem cell-like phenotypes of HCC cells. Representative photo-images (upper panel) and histograms (lower panel) comparing (**A**) migration and (**B**) invasion potential between Mahlavu and Mahlavu-R cells. Scale bar: 100 μm, 10× objective. (**C**) Representative photo-images (left panel) comparing clonogenicity between Huh7-R, Mahlavu-R, Hep3B-R and their wild-type counterparts. Scale bar: 100 μm, 10× objective. (**D**) Representative Western blot images of the differential expression of p-PDK1, PDK1, E-cadherin, *N*-cadherin, Vimentin or Snail protein in wild-type and IR-resistant Hep3B, Mahlavu or Huh7 cells. β-actin was used as a loading control. (**E**) Representative photo-images (upper panel) and histograms (lower panel) comparing the tumorsphere formation potential between wild-type (WT) and IR-resistant Huh7 or Mahlavu cells. Scale bar: 100 μm, 10× objective. (**F**) Representative immunofluorescence staining images of SOX2 and OCT4A expression in WT and IR-resistant Huh7 or Mahlavu cells. DAPI served as nuclear marker. Scale bar: 100 μm, 10× objective. * *p* < 0.05, ** *p* < 0.01, *** *p* < 0.001.

**Figure 5 cells-09-00746-f005:**
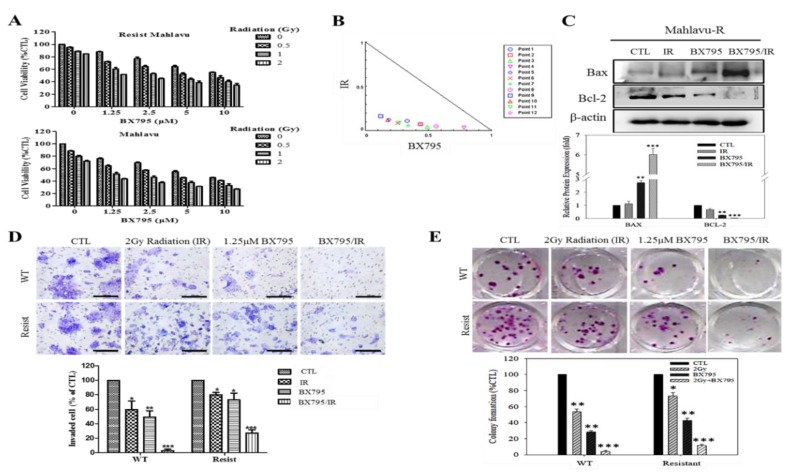
BX795-mediated pharmacologic targeting of PDK1 resensitizes HCC cells to radiotherapy-induced apoptosis signals dose-dependently and significantly suppresses their oncogenicity. (**A**) Histograms of the effect of 24 h exposure to 0.5–2 Gy IR on the viability of IR-resistant (upper panel) or wild-type (lower panel) Mahlavu cells. (**B**) Isobologram indicating synergism between different doses of IR and BX795. (**C**) The effect of exposure to 2 Gy IR, 1.25 μM BX795 or 2 Gy IR+1.25 μM BX795 on the expression level of Bax and Bcl-2 proteins in Mahlavu-R cells as shown by Western blot analyses. Photo-images (upper panel) and histograms (lower panel) comparing the effect of exposure to 2 Gy IR, 1.25 μM BX795 or 2 Gy IR+1.25 μM BX795 on the (**D**) invasion, and (**E**) colony formation potential of WT or IR-resistant Mahlavu cells. β-actin was used as a loading control. Scale bar: 100 μm, 10× objective; * *p* < 0.05, ** *p* < 0.01, *** *p* < 0.001; Resist, IR-resistant; WT, wild-type; CTL, control.

**Figure 6 cells-09-00746-f006:**
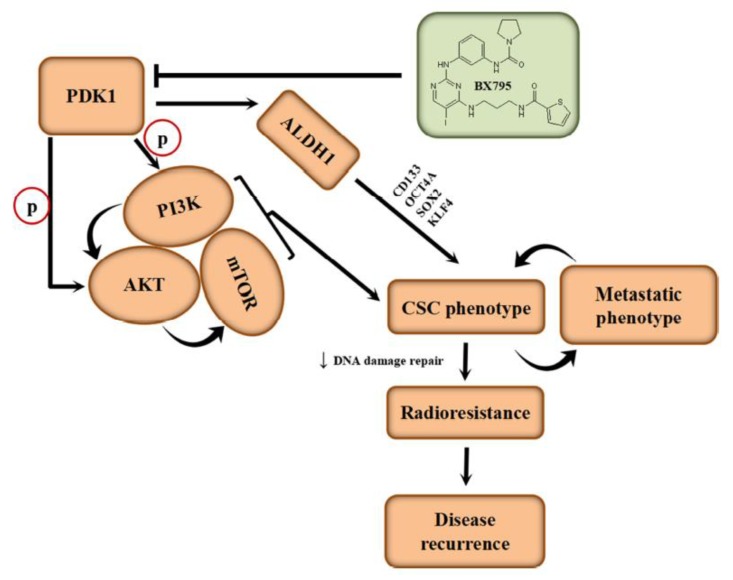
Schematic abstract of the aberrantly expressed PDK1 independently drives PI3K/AKT/mTOR oncogenic signaling in HCC. This present study demonstrated that aberrantly expressed PDK1 independently drives PI3K/AKT/mTOR oncogenic signaling, characterizes poorly differentiated cells, activates ALDH1 and is associated with the desensitization of aggressive hepatocellular carcinoma cells to radiotherapy.

## Data Availability

The datasets used and analyzed in the current study are publicly accessible as indicated in the manuscript.

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
