# Peer review of "Elevated PDK1 Expression Drives PI3K/AKT/MTOR Signaling Promotes Radiation-Resistant and Dedifferentiated Phenotype of Hepatocellular Carcinoma"

_cells, 2020, doi:10.3390/cells9030746_

Round 1
Reviewer 1 Report
The authors demonstrate novel findings for the role of PDK1 in HCC and radiation resistance. Please address following comments.
Please describe antibodies phosphorylation targets in materials and methods (for example, p-Akt (Ser473)) Correct in materials and methods and Fig. 1C pPI3K and PI3K to PI3 Kinase p85 Antibody. It’s a regulatory subunit expression and phosphorylation, not PI3K expression. Please clarify either in materials and methods or in manuscript text origin of E-GEOD-6465, A-AFFY-44, AFFY_HG_U133_PLUS datasets Line 449-450 please format formula photo to match text line height Please correct “µM” in Figure 5 legend as it became “1.25 @M”
Author Response
Response to Reviewers: Point-by-point responses to reviewer’s comments - Reviewer 1: We thank the reviewer for carefully reading our manuscript and providing valuable comments. We believe making use of all these comments has further helped improve the quality and appeal of our work, as well as strengthened the manuscript. Below are our point-by-point responses. Q1.1. The authors demonstrate novel findings for the role of PDK1 in HCC and radiation resistance. Please address following comments. A1.1. We thank the reviewer for taking time to read our manuscript and for the critiques and suggestions made in order to help us improve the quality of our work. In this revised manuscript, we have made effort to make address all the comments and suggestions. Q1.2. Please describe antibodies phosphorylation targets in materials and methods (for example, p-Akt (Ser473)) Correct in materials and methods and Fig. 1C pPI3K and PI3K to PI3 Kinase p85 Antibody. It’s a regulatory subunit expression and phosphorylation, not PI3K expression. A1.2. We sincerely thank the reviewer for these comments. We have now revised our manuscript to address these issues raised by the reviewer. Please kindly see our revised Materials & Methods section, Lines 181-201. 2.7. Western Blot Analysis After separation of 20 μg protein samples using 10% sodium dodecyl sulfate – polyacrylamide gel electrophoresis (SDS-PAGE) gel, the protein blots were transferred onto Polyvinylidene fluoride (PVDF) membranes in the Bio-Rad Mini-Protein electro-transfer system (Bio-Rad Laboratories, Inc, CA, USA), followed by membrane-blocking in 5% skimmed milk in Tris-buffered saline with Tween 20 (TBST) for 1 h. Thereafter, membranes were incubated overnight at 4◦C with primary antibodies against p-PI3 Kinase p85 (Tyr458)/p55 (Tyr199) (#4228S; 1 : 1000, Cell Signaling Technology Inc., Danvers, MA, USA), PI3 Kinase p110α (#4292S; 1 : 1000, Cell Signaling Technology), p-PDK1 (Ser241) (#3061L; 1 : 1000, Cell Signaling Technology), PDK1 (#3062S; 1 : 1000, Cell Signaling Technology), p-AKT (Ser473) (#9271L; 1 : 1000, Cell Signaling Technology), AKT (#4691L; 1 : 1000, Cell Signaling Technology), p-mTOR (Ser2448) (#2971L; 1 : 1000, Cell Signaling Technology), mTOR (#2972S; 1 : 1000, Cell Signaling Technology), E-cadherin (#3195S; 1 : 1000, Cell Signaling Technology), N-cadherin (#13116S; 1 : 1000, Cell Signaling Technology), Vimentin (#5741S; 1 : 1000, Cell Signaling Technology), Snail (#3879S; 1 : 1000, Cell Signaling Technology), Bcl-2 (#15071S; 1 : 1000, Cell Signaling Technology), Bax (#5023S 1 : 1000, Cell Signaling Technology), and β-actin (sc-69879; 1 : 500, Santa Cruz Biotechnology, Santa Cruz, CA, USA) in Supplementary Table S1. This was followed by incubation of membranes in appropriate secondary antibodies conjugated with horseradish peroxidise (HRP) at room temperature for 1 h, washed carefully with PBS thrice, and then protein band detection performed using the enhanced chemiluminescence (ECL) detection system (Thermo Fisher Scientific Inc, Waltham, MA, USA), and band densitometry-based quantification done using the ImageJ software (https://imagej.nih.gov/ij/). Please also kindly see our revised Results section, Lines 319~325. Furthermore, results of our western blot analyses showed that compared to its non-expression in the normal adult liver epithelial THLE-2 cells, the expression of p-PI3K p85(Tyr458)/p55(Tyr199), PI3K, p-PDK1 (Ser 241), PDK1, p-AKT (Ser 473), AKT, p-mTOR (Ser 2448), and mTOR proteins were highly or moderately-highly enhanced in the poorly-differentiated SK-HEP1 and Mahlavu, and Hep3B cell lines, moderately enhanced in poorly-differentiated FOCUS (CVCL_7955) cells, but mildly or not expressed in the well-differentiated Huh7 cells (Figure 1C). Also kindly see our revised Figure 1 legend, Lines 334-345. Figure 1. PDK1 is an independent driver of the PI3K/AKT/mTOR signaling pathway and its aberrant expression characterizes poorly differentiated aggressive HCC cells. (A) Box and whisker plots of the differential expression of PI3K, PDK1, AKT, and mTOR in HCC or normal liver tissues from the Wurmbach liver cohort. (B) STRINGdb-generated visualization of the protein-protein interaction between PDK1/PDPK1, and molecular components of the PI3K/AKT/mTOR signaling pathway. (C) Representative western blot images showing the differential expression of p-PI3K p85(Tyr458)/p55(Tyr199), PI3K, p-PDK1 (Ser 241), PDK1, p-AKT (Ser 473), AKT, p-mTOR (Ser 2448), and mTOR proteins in THLE-2, CVCL_7955, Mahlavu, Huh7, SK-HEP1 or Hep3B cell lines. (D) Representative IHC images of PDK1 immunoreactivity in Stage I – Stage IV HCC, compared to normal liver tissue. b-actin served as loading control. *p
Reviewer 2 Report
Lines 91-93 (This is not a complete sentence. Please see recommendation.)
Thus, the need for continued unraveling of the mechanistic underlining of IR-resistance in HCC, identification of reliable molecular targets, and development of more effective therapies may be warranted.
Line 255 (A space is needed between with10%).
Author Response
Point-by-point responses to reviewer’s comments - Reviewer 2:
We would like to thank the reviewer for the thorough reading of our manuscript as well as their valuable comments. We believe all comments are borne out of good faith, and thus, have tried to address their comments conscientiously and feel that they have further improved the readability and appeal of our work, as well as strengthened the manuscript. Below are our point-by-point responses.
Q2.1. Lines 91-93 (This is not a complete sentence. Please see recommendation.) Thus, the need for continued unraveling of the mechanistic underlining of IR-resistance in HCC, identification of reliable molecular targets, and development of more effective therapies may be warranted.
A2.1. We sincerely appreciate the reviewer’s comments. Please kindly see our revised Background section, Lines 90-96.
The role of IR in the treatment of HCC continues to evolve, especially with regards to technological advancement and in the context of combinatorial therapy, aimed at enhancing IR safety and efficacy; nevertheless, the anti-HCC efficacy of IR is non-apparent in the intrinsically IR-resistant cells or blunted over time by the acquisition of IR-resistance, subsequently resulting in disease relapse and poor prognosis [7, 8]. Thus, the need for continued unraveling of the mechanistic underlining of IR-resistance in HCC, identification of reliable molecular targets, and development of more effective therapies.
Q2.3. Line 255 (A space is needed between with10%).
A2.3. We thank the reviewer for the keen observation. As suggested by the reviewer, we have now inserted space between ‘with’ and ‘10%’ in our revised manuscript. Please kindly see our revised Materials and Methods section, Lines 255-262.
2.13. Scratch-Wound Healing Migration Assay
To evaluate cell migration, we used well-established protocol. Briefly, Mahlavu, or Mahlavu-R cells were seeded into 6-well plates (Corning Inc., Corning, NY, USA) containing complete growth media with 10% FBS, cultured to 98–100% confluence, then the median axes of the cell monolayers were denuded with sterile yellow pipette tips. The scratch-wound healing cum cell migration was monitored over time and images captured under a light microscope with 10× objective lens at the 0 and 48 h time-points after denudation, and then images were analyzed with the NIH ImageJ software (https://imagej.nih.gov/ij/download.html).

Reviewer 3 Report
This study determined the roles of PDK1 in radiation-resistance and dedifferentiated phenotype of hepatocellular carcinoma via a numerous assays, including bioinformatics analyses, functional protein–protein interaction analysis and other combined cell biological assays. The authors draw the conclusions that PDK1 is an active driver of IR resistance by activation of the PI3K/AKT/mTOR signaling, up-modulation of cancer stemness signaling, and suppression of DNA damage and targeting PDK1 is a potential new therapeutic approach for treating IR-resistant HCC. Although the results presented in this manuscript provide a certain level of advance in current knowledge, this is a multiple focus study without a precise hypothesis. In addition, the most conclusions are obtained from the results of correlation and association studies. That being said, the conclusions may be interesting for the readership of the Journal.
The following suggestion may be considered:
1) What is the relation between cancer stemness signaling, and suppression of DNA damage, metastatic phenotype and augmented EMT? Are those phenotypes related to PI3K/AKT/mTOR pathways?
2) The data on metastatic phenotype and augmented EMT do not contribute much to the main conclusions. In contrast, it is a kind of distraction.
3) The roles of ALDH1 and PI3K pathway in PDK1-mediated radioresistance should be verified in the cells without ALDH1 expression or in the cells with deficient PI3K/AKT/mTOR, respectively.
4) According to the model in Figure 6, ALDH1 and PI3K/AKT/mTOR pathways are parallel. Dose PDK1 knockdown still sensitize the cells to IR in the cells without ALDH1 or in the cells with a deficiency of PI3K/ATK/mTOR pathway?
5)There are some issue with rigor of science, For instance, in Fig.2D, p-AKT level appears to be similar between group of siPDK and 2GY+siPDK1; in Fig.1D and Fig.3C,F, only representative data are shown, no statistical analysis are performed.
6) It is not clear why AKT mRNA is downregulated but the PI3K and mTOR are upregulated (Figure 1A).
7) If PDK1 expression contributes to radioresistance, why downregulated expression of DDR genes Rad50, MSH3, ERCC2 and BLM gene cluster were observed (Fig.3A). Lower expression of DDR genes should lead to the sensitivity to radiotherapy.
8) The protein-protein interaction of ALDH1 and PDK1 is not validated by co-IP and protein-protein interaction assay or other biological assays.
Author Response
Point-by-point responses to reviewer’s comments - Reviewer 3:
We would like to thank the reviewer for the thorough reading of our manuscript as well as their valuable comments. We believe all comments are borne out of good faith, and thus, have tried to address their comments conscientiously and feel that they have further improved the readability and appeal of our work, as well as strengthened the manuscript. Below are our point-by-point responses.
Q3.1. This study determined the roles of PDK1 in radiation-resistance and dedifferentiated phenotype of hepatocellular carcinoma via a numerous assay, including bioinformatics analyses, functional protein–protein interaction analysis and other combined cell biological assays. The authors draw the conclusions that PDK1 is an active driver of IR resistance by activation of the PI3K/AKT/mTOR signaling, up-modulation of cancer stemness signaling, and suppression of DNA damage and targeting PDK1 is a potential new therapeutic approach for treating IR-resistant HCC. Although the results presented in this manuscript provide a certain level of advance in current knowledge, this is a multiple focus study without a precise hypothesis. In addition, the most conclusions are obtained from the results of correlation and association studies. That being said, the conclusions may be interesting for the readership of the Journal.
A3.1. We are very grateful to the reviewer for taking time to read our manuscript and for the critiques and suggestions made in order to help us improve the quality of our work. In this revised manuscript, we have made effort to address all the comments and suggestions.
Q3.2. 1) What is the relation between cancer stemness signaling, and suppression of DNA damage, metastatic phenotype and augmented EMT? Are those phenotypes related to PI3K/AKT/mTOR pathways?
A3.2. We sincerely appreciate the reviewer’s comment. We apologize that the erudite reviewer must have missed this while reading our initial submission. Please kindly see our revised Results section, Lines 306-332.
3.1. PDK1 is an independent driver of the PI3K/AKT/mTOR signaling pathway and its aberrant expression characterizes poorly differentiated aggressive HCC cells.
Following our in-house mining of high through-put gene expression data from public repositories, to unravel the role of the PI3K/AKT/mTOR signaling axis, and more specifically PDK1 signaling in HCC, we performed in silico proteo-transcriptomic analyses of selected molecular components of the pathway in question. Results of our differential expression profiling of the Wurmbach liver cohort (n = 75), with 45 context-relevant sample, revealed concomitant significant up-regulation of PI3K (1.22-fold, p = 0.030), PDK1 (2.17-fold, p = 8.95x10-6), mTOR (1.37-fold, p = 6.26x10-4), and statistically insignificant down-regulation of AKT mRNA (-1.14-fold, p = 0.847) in HCC samples (n = 35) compared to the normal liver tissues (n = 10) (Figure 1A). Protein-protein interaction (PPI) enrichment analyses using the STRING-db (https://string-db.org) platform for PPI network prediction revealed very strong association between PI3K, PDK1, AKT, mTOR, and other indicated effectors and/or mediators of the PI3K/AKT/mTOR signaling axis, as expressed by the relatively high mean local clustering coefficient of 88.3% (p < 1.0x10-16 (Figure 1B). Furthermore, results of our western blot analyses showed that compared to its non-expression in the normal adult liver epithelial THLE-2 cells, the expression of p-PI3K p85(Tyr458)/p55(Tyr199), PI3K, p-PDK1 (Ser 241), PDK1, p-AKT (Ser 473), AKT, p-mTOR (Ser 2448), and mTOR proteins were highly or moderately-highly enhanced in the poorly-differentiated SK-HEP1 and Mahlavu, and Hep3B cell lines, moderately enhanced in poorly-differentiated FOCUS (CVCL_7955) cells, but mildly or not expressed in the well-differentiated Huh7 cells (Figure 1C). Moreover, coupled with the consistent high mRNA intensity and protein expression levels of PDK1 in earlier data, from IHC-based proteome analyses using the human protein atlas platform version 18.1 (https://www.proteinatlas.org/ENSG00000152256-PDK1/pathology/tissue/liver+cancer#img) we observed a significant positive correlation between PDK1 immunoreactivity/expression and disease stage/progression (Figure 1D). These data do indicate, at least in part, that PDK1 is an independent driver of the PI3K/AKT/mTOR signaling pathway and its aberrant expression characterizes poorly differentiated aggressive HCC cells (Figure 1D).
Please also kindly see our revised Results section, Lines 346-369.
3.2 Altered PDK1 expression is sufficient for the deactivation of the PI3K/PDK/AKT/mTOR oncogenic signaling and sensitizes aggressive HCC cells to radiotherapy.
Having shown that enhanced PDK1 expression plays an essential role in the PI3K/AKT/mTOR signaling axis and characterizes poorly differentiated aggressive HCC cells, to gain some insight into the mechanistic underlining of PDK1 on HCC oncogenicity and therapy response, we evaluated the effect of increasing doses of IR on different HCC cells. We observed that while 24 h exposure to 0.5 Gy - 2 Gy IR significantly reduced the viability of PDK1low Huh7 (46.1%, p < 0.05 at 2Gy) and PDK1low/moderate FOCUS cells (38.4%, p < 0.05 at 2Gy), its effect on the PDK1moderate Hep3B cell was mild (~24% at 2Gy), and it had no apparent or insignificant cytocidal effect on the PDK1high SK-HEP1 and Mahlavu cells, respectively (Figure 2A). Furthermore, we demonstrated that aside eliciting a 59% decline in the viability of PDK1high Mahlavu cells, siRNA-mediated loss-of-PDK1 function (siPDK1) enhanced their sensitivity to 2Gy IR by a significant 80%; similarly in synergism with siPDK1, the cytocidal effect of 2Gy IR was increased to 97% from 42% alone in PDK1low Huh7 cells, and to 93% from 26% alone in PDK1moderate Hep3B cells (Figure 2B), thus, indicating a vital role for PDK1 expression and/or activity in the radiosensitivity of HCC cells. In addition, compared to the 27%, 14% or 51% loss of clonogenicity elicited by exposure to 2Gy IR in Mahlavu, Hep3B, or Huh7 cells, respectively, siPDK1 inhibited the ability of Mahlavu, Hep3B, or Huh7 cells to form colonies by 91%, 96%, or 99.8%, respectively (Figure 2C). Combining siPDK1 with 2Gy IR was incompatible with clonal survival in all three cell lines (Data not provided). Interestingly, the observed inhibition of cell viability and attenuation of clonogenicity by siPDK1 alone or in synergism with 2Gy IR, were associated with down-regulated p-PDK1, PDK1, p-PI3K p85(Tyr458)/p55(Tyr199), PI3K, p-AKT, AKT, p-mTOR, and mTOR in the PDK1high Mahlavu cells (Figure 2D). These results do indicate that altered PDK1 expression via specific targeting of PDK1 is sufficient to deactivate the PI3K/AKT/mTOR oncogenic signaling in HCC cells and sensitizes the aggressive cells to IR.
We also believe all these is succinctly put in our Discussion section.
Q3.3. 2) The data on metastatic phenotype and augmented EMT do not contribute much to the main conclusions. In contrast, it is a kind of distraction.
A3.3. We thank the reviewer for this insightful comment. We have however included this as is customary with studies of this kind considering the connection between the cancer stemness, therapy failure, EMT, enhanced metastasis, and disease recurrence. Nevertheless, the principal focus of the present work remains that “aberrantly expressed PDK1 independently drives PI3K/AKT/mTOR oncogenic signaling, characterizes poorly differentiated cells, activates ALDH1, and is associated with the desensitization of aggressive Hepatocellular Carcinoma cells to radiotherapy.”
Please kindly see our revised Conclusion section, Lines 636-644.
- Conclusion
This present study as we depicted in our schematic abstract (Figure 6), demonstrated that aberrantly expressed PDK1 independently drives PI3K/AKT/mTOR oncogenic signaling, characterizes poorly differentiated cells, activates ALDH1, and is associated with the desensitization of aggressive Hepatocellular Carcinoma cells to radiotherapy. Thus, we provide preclinical evidence for the pleiotropic contribution of PDK1 to the IR-resistant phenotype of aggressive HCC cells by modulating oncogenic, stemness, metastatic and DNA damage repair signaling. These findings provide important mechanistic insights into HCC biology and have significant therapeutic implications.
Q3.4. 3) The roles of ALDH1 and PI3K pathway in PDK1-mediated radioresistance should be verified in the cells without ALDH1 expression or in the cells with deficient PI3K/AKT/mTOR, respectively.
A3.4. We are very grateful for the reviewer’s comments. We have now revised our manuscript consistent with the Reviewer’s concern. Please kindly see our revised Results section, Lines 381-424.
3.1. PDK1 is an independent driver of the PI3K/AKT/mTOR signaling pathway and its aberrant expression characterizes poorly differentiated aggressive HCC cells.
Following our in-house mining of high through-put gene expression data from public repositories, to unravel the role of the PI3K/AKT/mTOR signaling axis, and more specifically PDK1 signaling in HCC, we performed in silico proteo-transcriptomic analyses of selected molecular components of the pathway in question. Results of our differential expression profiling of the Wurmbach liver cohort (n = 75), with 45 context-relevant sample, revealed concomitant significant up-regulation of PI3K (1.22-fold, p = 0.030), PDK1 (2.17-fold, p = 8.95x10-6), mTOR (1.37-fold, p = 6.26x10-4), and statistically insignificant down-regulation of AKT mRNA (-1.14-fold, p = 0.847) in HCC samples (n = 35) compared to the normal liver tissues (n = 10) (Figure 1A). Protein-protein interaction (PPI) enrichment analyses using the STRING-db (https://string-db.org) platform for PPI network prediction revealed very strong association between PI3K, PDK1, AKT, mTOR, and other indicated effectors and/or mediators of the PI3K/AKT/mTOR signaling axis, as expressed by the relatively high mean local clustering coefficient of 88.3% (p < 1.0x10-16 (Figure 1B). Furthermore, results of our western blot analyses showed that compared to its non-expression in the normal adult liver epithelial THLE-2 cells, the expression of p-PI3K p85(Tyr458)/p55(Tyr199), PI3K, p-PDK1 (Ser 241), PDK1, p-AKT (Ser 473), AKT, p-mTOR (Ser 2448), and mTOR proteins were highly or moderately-highly enhanced in the poorly-differentiated SK-HEP1 and Mahlavu, and Hep3B cell lines, moderately enhanced in poorly-differentiated FOCUS (CVCL_7955) cells, but mildly or not expressed in the well-differentiated Huh7 cells (Figure 1C). Moreover, coupled with the consistent high mRNA intensity and protein expression levels of PDK1 in earlier data, from IHC-based proteome analyses using the human protein atlas platform version 18.1 (https://www.proteinatlas.org/ENSG00000152256-PDK1/pathology/tissue/liver+cancer#img) we observed a significant positive correlation between PDK1 immunoreactivity/expression and disease stage/progression (Figure 1D). These data do indicate, at least in part, that PDK1 is an independent driver of the PI3K/AKT/mTOR signaling pathway and its aberrant expression characterizes poorly differentiated aggressive HCC cells (Figure 1D).
Please also kindly see our revised Results section, Lines 346-369.
3.2 Altered PDK1 expression is sufficient for the deactivation of the PI3K/PDK/AKT/mTOR oncogenic signaling and sensitizes aggressive HCC cells to radiotherapy.
Having shown that enhanced PDK1 expression plays an essential role in the PI3K/AKT/mTOR signaling axis and characterizes poorly differentiated aggressive HCC cells, to gain some insight into the mechanistic underlining of PDK1 on HCC oncogenicity and therapy response, we evaluated the effect of increasing doses of IR on different HCC cells. We observed that while 24 h exposure to 0.5 Gy - 2 Gy IR significantly reduced the viability of PDK1low Huh7 (46.1%, p < 0.05 at 2Gy) and PDK1low/moderate FOCUS cells (38.4%, p < 0.05 at 2Gy), its effect on the PDK1moderate Hep3B cell was mild (~24% at 2Gy), and it had no apparent or insignificant cytocidal effect on the PDK1high SK-HEP1 and Mahlavu cells, respectively (Figure 2A). Furthermore, we demonstrated that aside eliciting a 59% decline in the viability of PDK1high Mahlavu cells, siRNA-mediated loss-of-PDK1 function (siPDK1) enhanced their sensitivity to 2Gy IR by a significant 80%; similarly in synergism with siPDK1, the cytocidal effect of 2Gy IR was increased to 97% from 42% alone in PDK1low Huh7 cells, and to 93% from 26% alone in PDK1moderate Hep3B cells (Figure 2B), thus, indicating a vital role for PDK1 expression and/or activity in the radiosensitivity of HCC cells. In addition, compared to the 27%, 14% or 51% loss of clonogenicity elicited by exposure to 2Gy IR in Mahlavu, Hep3B, or Huh7 cells, respectively, siPDK1 inhibited the ability of Mahlavu, Hep3B, or Huh7 cells to form colonies by 91%, 96%, or 99.8%, respectively (Figure 2C). Combining siPDK1 with 2Gy IR was incompatible with clonal survival in all three cell lines (Data not provided). Interestingly, the observed inhibition of cell viability and attenuation of clonogenicity by siPDK1 alone or in synergism with 2Gy IR, were associated with down-regulated p-PDK1, PDK1, p-PI3K p85(Tyr458)/p55(Tyr199), PI3K, p-AKT, AKT, p-mTOR, and mTOR in the PDK1high Mahlavu cells (Figure 2D). These results do indicate that altered PDK1 expression via specific targeting of PDK1 is sufficient to deactivate the PI3K/AKT/mTOR oncogenic signaling in HCC cells and sensitizes the aggressive cells to IR.
We also believe all these is succinctly put in our Discussion section.
Q3.5. According to the model in Figure 6, ALDH1 and PI3K/AKT/mTOR pathways are parallel. Dose PDK1 knockdown still sensitize the cells to IR in the cells without ALDH1 or in the cells with a deficiency of PI3K/ATK/mTOR pathway?
A3.5. We thank the reviewer for these comments. While we are not sure we fully understand the reviewer’s question, we do believe that we have provided data indicating the cross-talk between PDK1 and ALDH1 and their role in the modulation of liver cancer cell sensitivity to IR. May we politely refer the reviewer to our revised Results section, Lines 381-424.
3.3 Aberrant PDK1 expression is implicated in the acquisition of radioresistance and evasion of DNA damage by HCC cells.
Because of the implication of DNA damage in the death of cancerous cells exposed to IR and the therapeutic benefit of exploiting the reduced resolution of IR-induced clustered DNA damage [16] and altered DNA damage repair (DDR) genes in liver cancer [17], firstly, we probed and re-analyzed Huynh H et al’s E-GEOD-6465, A-AFFY-44, AFFY_HG_U133_PLUS_2 data set on the array expression profiling of xenografts of HCC (n = 53 samples, 54675 genes) (https://www.ebi.ac.uk/arrayexpress/experiments/E-GEOD-6465/). Generated expression-based heat-map revealed a dichotomization of our selected gene-sets, such that the PDK1, ALDH1A1, CD133/PROM1, OCT4A/POU5F1, SOX2, and KLF4 genes clustering with TP53 were up-regulated, while the DDR genes RAD50, MSH3, MLH3, ERCC2, and BLM gene cluster were down-regulated (Figure 3A). Additional analyses using the STRING-db platform (https://string-db.org) for PPI network prediction further confirmed earlier results, as we observed a very strong association between components of the PI3K/AKT signaling, namely PDK1/PDPK1, AKT1, MTOR, and stemness marker complicit in IR-resistance ALDH1Al, CD133/PROM1, OCT4A/POU5F1, SOX2, and KLF4 pooled together, while DNA damage markers RAD50, MSH3, MLH3, ERCC2, and BLM pooled together (Figure 3B). The average local clustering coefficient for the clustered proteins was 0.827 and PPI enrichment p-value was p < 2.45 x 10-10 (Figure 3B). Because of the suggested implication of cancer stem cell (CSCs) markers in PDK1-induced IR-resistance, we further examined if and to what extent PDK1-induced IR-resistance affects the side population (SP) which is representative of the CSCs pool in vitro. Comparative analyses of HCC wild type and PDK1-rich IR-resistant clones (HCC-R) revealed a 1.40-fold increase in the SP in Mahlavu-R compared to its wild type counterpart; similarly compared to Hep3B or Huh7 cells, the SP was increased by 2.14-fold or 7.03-fold in Hep3B-R or Huh7-R cells, respectively (Figure 3C). More so, because of the documented implication of ALDH1 in IR-resistance [18], we probed the GDC TCGA liver cancer (LIHC, n = 469) for probable relationship between PDK1 and ALDH, and showed a positive correlation between ALDH1A1 and PDK1 (R =0.27, p = 1.6 x 10-8) (Figure 3D). Consistent with the ‘co-expression - function similarity’ paradigm, and in conformity with conventional knowledge that when an inactive enzyme, otherwise known as ‘apoenzyme’, (in this case, apoALDH1A1) binds with an organic or inorganic helper-molecule/co-factor, a complete and catalytically active form of the enzyme called an ‘holoenzyme’ is formed, we generated a spatiotemporal visualization of the probable interaction between PDK1 and ALDH1 using the Schrödinger’s PyMOL molecular graphics system (https://pymol.org/2/), and demonstrated that the catalytic domain of PDK1 (protein data bank, PDB: 1H1W) binds directly with human apoALDH1A1 (PDB: 4WJ9) with an interaction score of 17.5, an atomic contact energy (ACE) of -50.05 kcal/mol, and root-mean-square deviation (RMSD) of 23.52Å (Figure 3E, also see Supplementary Figure S1). In parallel assays, we observed significantly enhanced ALDH1 activity in the PDK1-rich Huh7-R cells, as demonstrated by a 24.16-fold increase in ALDH activity in the erstwhile PDK1low Huh7 cells when they acquire an IR-resistant phenotype (Huh7-R) (Figure 3F), which is consistent with our predicted PDK1-ALDH1 interaction probability of 0.80 or 0.99 using the random forest (RF) or support-vector machine (SVM) classifier algorithm, respectively, as shown in Figure 3D. PDK1 interacts with ALDH and directly modulate the expression and/or activity of ALDH in HCC cells. Representative western blot image and histograms of the differential expression of PDK1 and ALDH1 in adherent wild-type Mahlavu cells or their tumorsphere counterparts (Supplementary Figure S2). These data indicate, at least in part, that PDK1 directly interacts with and activates ALDH1A1, and implicates the aberrant PDK1 expression in the acquisition of IR-resistance and evasion of DNA damage by HCC cells.
Q3.6. 5)There are some issue with rigor of science, For instance, in Fig.2D, p-AKT level appears to be similar between group of siPDK and 2GY+siPDK1; in Fig.1C and Fig.3C,F, only representative data are shown, no statistical analysis are performed.
A3.6. We are grateful for the reviewer’s comment. We have now addressed the issues raised by reviewer by including graphical representation of the data in question. Please see the updated figure 1 and figure 2. Also kindly see our updated Figure 1 and Figure 2 legend, Lines 334-345 and line 371-380.
Figure 1. PDK1 is an independent driver of the PI3K/AKT/mTOR signaling pathway and its aberrant expression characterizes poorly differentiated aggressive HCC cells. (A) Box and whisker plots of the differential expression of PI3K, PDK1, AKT, and mTOR in HCC or normal liver tissues from the Wurmbach liver cohort. (B) STRINGdb-generated visualization of the protein-protein interaction between PDK1/PDPK1, and molecular components of the PI3K/AKT/mTOR signaling pathway. (C) Representative western blot images and histograms showing the differential expression of p-PI3K p85(Tyr458)/p55(Tyr199), PI3K, p-PDK1 (Ser 241), PDK1, p-AKT (Ser 473), AKT, p-mTOR (Ser 2448), and mTOR proteins in THLE-2, CVCL_7955, Mahlavu, Huh7, SK-HEP1 or Hep3B cell lines. (D) Representative IHC images of PDK1 immunoreactivity in Stage I – Stage IV HCC, compared to normal liver tissue. β-actin served as loading control. *p<0.05, @p<0.01, #p<0.001 vs THLE-2; Green nodes, PI3K/AKT signaling; Red nodes, mTOR signaling; PPI, protein-protein interaction.
Also kindly see our updated Figure 2D and its legend, Page 9, Lines 372-383:
Figure 2. Altered PDK1 expression is enough for the deactivation of the PI3K/PDK/AKT/mTOR oncogenic signaling and sensitizes aggressive HCC cells to radiotherapy. (A) Histograms of the effect of 24 h exposure to 0.5Gy – 2Gy IR on the viability of SK-HEP1, Huh7, Hep3B, CVCL_7955 or Mahlavu cell lines. (B) Representative photo-images of the effect of 2Gy, siPDK1 or 2Gy+siPDK1 on the viability/proliferation, and morphology of Mahlavu, Huh7 or Hep3B cells. (C) Representative photo-images showing the effect of 2Gy or siPDK1 on clonogenicity of Mahlavu, Huh7 or Hep3B cells. (D) Representative western blot images and histograms showing the effect of 2Gy, siPDK1 or 2Gy+siPDK1 on the expression levels of p-PI3K p85(Tyr458)/p55(Tyr199), PI3K, p-PDK1 (Ser 241), PDK1, p-AKT (Ser 473), AKT, p-mTOR (Ser 2448), and mTOR proteins in Mahlavu cells. *p<0.05, @p<0.01, #p<0.001 vs CTL; ns, not significant.
Q3.7. 6) It is not clear why AKT mRNA is downregulated but the PI3K and mTOR are upregulated (Figure 1A).
A3.7. We appreciate the reviewer’s comment. It is unclear to us as well “why AKT mRNA is downregulated but the PI3K and mTOR are upregulated”, however, it does make sense considering that this expression profile of PI3K, AKT, and mTOR transcripts are from clinical samples, especially in the context of tissue peculiarity and specificity. Indeed, we believe it is not unknown to the erudite reviewer that “pathways are deferentially expressed between cell lines and their tissue origin.” [Lopes-Ramos CM, et al. Regulatory network changes between cell lines and their tissues of origin. BMC Genomics. 2017; 18:723], and “while specificity is often described based on gene expression levels,…by themselves, individual genes, or even sets of genes, cannot adequately capture the variety of processes that distinguish different tissues.” [Sonawane AR, et al. Understanding Tissue-Specific Gene Regulation. Cell Reports. 2017; 21(4):1077-1088]. May we also humbly refer the reviewer to Lukk M, et al. A global map of gene expression. Nat Biotechnol. 2010; 28: 322-324.
Q3.8. 7) If PDK1 expression contributes to radioresistance, why downregulated expression of DDR genes Rad50, MSH3, ERCC2 and BLM gene cluster were observed (Fig.3A). Lower expression of DDR genes should lead to the sensitivity to radiotherapy.
A3.8. We thank the reviewer for this comment. Yes, we do agree with the reviewer that lower expression of DNA damage repair/response genes elicits increased sensitivity to radiotherapy. This is consistent with our results showing that while PDK1 clusters with markers of stemness and resistance to therapy, the DDR proteins form a separate cluster of their own. Please kindly see our revised manuscript, Results section, Lines 381-424.
3.3 Aberrant PDK1 expression is implicated in the acquisition of radioresistance and evasion of DNA damage by HCC cells.
Because of the implication of DNA damage in the death of cancerous cells exposed to IR and the therapeutic benefit of exploiting the reduced resolution of IR-induced clustered DNA damage [16] and altered DNA damage repair (DDR) genes in liver cancer [17], firstly, we probed and re-analyzed Huynh H et al’s E-GEOD-6465, A-AFFY-44, AFFY_HG_U133_PLUS_2 data set on the array expression profiling of xenografts of HCC (n = 53 samples, 54675 genes) (https://www.ebi.ac.uk/arrayexpress/experiments/E-GEOD-6465/). Generated expression-based heat-map revealed a dichotomization of our selected gene-sets, such that the PDK1, ALDH1A1, CD133/PROM1, OCT4A/POU5F1, SOX2, and KLF4 genes clustering with TP53 were up-regulated, while the DDR genes RAD50, MSH3, MLH3, ERCC2, and BLM gene cluster were down-regulated (Figure 3A). Additional analyses using the STRING-db platform (https://string-db.org) for PPI network prediction further confirmed earlier results, as we observed a very strong association between components of the PI3K/AKT signaling, namely PDK1/PDPK1, AKT1, MTOR, and stemness marker complicit in IR-resistance ALDH1Al, CD133/PROM1, OCT4A/POU5F1, SOX2, and KLF4 pooled together, while DNA damage markers RAD50, MSH3, MLH3, ERCC2, and BLM pooled together (Figure 3B). The average local clustering coefficient for the clustered proteins was 0.827 and PPI enrichment p-value was p < 2.45 x 10-10 (Figure 3B). Because of the suggested implication of cancer stem cell (CSCs) markers in PDK1-induced IR-resistance, we further examined if and to what extent PDK1-induced IR-resistance affects the side population (SP) which is representative of the CSCs pool in vitro. Comparative analyses of HCC wild type and PDK1-rich IR-resistant clones (HCC-R) revealed a 1.40-fold increase in the SP in Mahlavu-R compared to its wild type counterpart; similarly compared to Hep3B or Huh7 cells, the SP was increased by 2.14-fold or 7.03-fold in Hep3B-R or Huh7-R cells, respectively (Figure 3C). More so, because of the documented implication of ALDH1 in IR-resistance [18], we probed the GDC TCGA liver cancer (LIHC, n = 469) for probable relationship between PDK1 and ALDH, and showed a positive correlation between ALDH1A1 and PDK1 (R =0.27, p = 1.6 x 10-8) (Figure 3D). Consistent with the ‘co-expression - function similarity’ paradigm, and in conformity with conventional knowledge that when an inactive enzyme, otherwise known as ‘apoenzyme’, (in this case, apoALDH1A1) binds with an organic or inorganic helper-molecule/co-factor, a complete and catalytically active form of the enzyme called an ‘holoenzyme’ is formed, we generated a spatiotemporal visualization of the probable interaction between PDK1 and ALDH1 using the Schrödinger’s PyMOL molecular graphics system (https://pymol.org/2/), and demonstrated that the catalytic domain of PDK1 (protein data bank, PDB: 1H1W) binds directly with human apoALDH1A1 (PDB: 4WJ9) with an interaction score of 17.5, an atomic contact energy (ACE) of -50.05 kcal/mol, and root-mean-square deviation (RMSD) of 23.52Å (Figure 3E, also see Supplementary Figure S1). In parallel assays, we observed significantly enhanced ALDH1 activity in the PDK1-rich Huh7-R cells, as demonstrated by a 24.16-fold increase in ALDH activity in the erstwhile PDK1low Huh7 cells when they acquire an IR-resistant phenotype (Huh7-R) (Figure 3F), which is consistent with our predicted PDK1-ALDH1 interaction probability of 0.80 or 0.99 using the random forest (RF) or support-vector machine (SVM) classifier algorithm, respectively, as shown in Figure 3D. PDK1 interacts with ALDH and directly modulate the expression and/or activity of ALDH in HCC cells. Representative western blot image and histograms of the differential expression of PDK1 and ALDH1 in adherent wild-type Mahlavu cells or their tumorsphere counterparts (Supplementary Figure S2). These data indicate, at least in part, that PDK1 directly interacts with and activates ALDH1A1, and implicates the aberrant PDK1 expression in the acquisition of IR-resistance and evasion of DNA damage by HCC cells.
Similarly, our gene expression heatmap showed that while PDK1 with the markers of stemness and resistance to therapy were upregulated, the DDR proteins were downregulated, and vice versa. Our data is consistent with contemporary evidence as can be seen in the following.
- Bhattacharya S, Asaithamby A. Repurposing DNA repair factors to eradicate tumor cells upon radiotherapy. Transl Cancer Res. 2017; 6(Suppl 5): S822–S839.
- Goldstein M, Kastan MB. The DNA damage response: implications for tumor responses to radiation and chemotherapy. Annu Rev Med. 2015; 66:129-43.
- Kinsella TJ. Understanding DNA Damage Response and DNA Repair Pathways: Applications to More Targeted Cancer Therapeutics. Semin Oncol. 2009; 36(2 Suppl 1): S42-51
Q3.9. 8) The protein-protein interaction of ALDH1 and PDK1 is not validated by co-IP and protein-protein interaction assay or other biological assays.
A3.9. We sincerely thank the reviewer for this insightful comment. As suggested by the reviewer, we have now included additional data corroborating the protein-to-protein interaction of ALDH1 and PDK1 using biological assays. Please kindly see our revised manuscript, Lines 381-424.
3.3 Aberrant PDK1 expression is implicated in the acquisition of radioresistance and evasion of DNA damage by HCC cells.
Because of the implication of DNA damage in the death of cancerous cells exposed to IR and the therapeutic benefit of exploiting the reduced resolution of IR-induced clustered DNA damage [16] and altered DNA damage repair (DDR) genes in liver cancer [17], firstly, we probed and re-analyzed Huynh H et al’s E-GEOD-6465, A-AFFY-44, AFFY_HG_U133_PLUS_2 data set on the array expression profiling of xenografts of HCC (n = 53 samples, 54675 genes) (https://www.ebi.ac.uk/arrayexpress/experiments/E-GEOD-6465/). Generated expression-based heat-map revealed a dichotomization of our selected gene-sets, such that the PDK1, ALDH1A1, CD133/PROM1, OCT4A/POU5F1, SOX2, and KLF4 genes clustering with TP53 were up-regulated, while the DDR genes RAD50, MSH3, MLH3, ERCC2, and BLM gene cluster were down-regulated (Figure 3A). Additional analyses using the STRING-db platform (https://string-db.org) for PPI network prediction further confirmed earlier results, as we observed a very strong association between components of the PI3K/AKT signaling, namely PDK1/PDPK1, AKT1, MTOR, and stemness marker complicit in IR-resistance ALDH1Al, CD133/PROM1, OCT4A/POU5F1, SOX2, and KLF4 pooled together, while DNA damage markers RAD50, MSH3, MLH3, ERCC2, and BLM pooled together (Figure 3B). The average local clustering coefficient for the clustered proteins was 0.827 and PPI enrichment p-value was p < 2.45 x 10-10 (Figure 3B). Because of the suggested implication of cancer stem cell (CSCs) markers in PDK1-induced IR-resistance, we further examined if and to what extent PDK1-induced IR-resistance affects the side population (SP) which is representative of the CSCs pool in vitro. Comparative analyses of HCC wild type and PDK1-rich IR-resistant clones (HCC-R) revealed a 1.40-fold increase in the SP in Mahlavu-R compared to its wild type counterpart; similarly compared to Hep3B or Huh7 cells, the SP was increased by 2.14-fold or 7.03-fold in Hep3B-R or Huh7-R cells, respectively (Figure 3C). More so, because of the documented implication of ALDH1 in IR-resistance [18], we probed the GDC TCGA liver cancer (LIHC, n = 469) for probable relationship between PDK1 and ALDH, and showed a positive correlation between ALDH1A1 and PDK1 (R =0.27, p = 1.6 x 10-8) (Figure 3D). Consistent with the ‘co-expression - function similarity’ paradigm, and in conformity with conventional knowledge that when an inactive enzyme, otherwise known as ‘apoenzyme’, (in this case, apoALDH1A1) binds with an organic or inorganic helper-molecule/co-factor, a complete and catalytically active form of the enzyme called an ‘holoenzyme’ is formed, we generated a spatiotemporal visualization of the probable interaction between PDK1 and ALDH1 using the Schrödinger’s PyMOL molecular graphics system (https://pymol.org/2/), and demonstrated that the catalytic domain of PDK1 (protein data bank, PDB: 1H1W) binds directly with human apoALDH1A1 (PDB: 4WJ9) with an interaction score of 17.5, an atomic contact energy (ACE) of -50.05 kcal/mol, and root-mean-square deviation (RMSD) of 23.52Å (Figure 3E, also see Supplementary Figure S1). In parallel assays, we observed significantly enhanced ALDH1 activity in the PDK1-rich Huh7-R cells, as demonstrated by a 24.16-fold increase in ALDH activity in the erstwhile PDK1low Huh7 cells when they acquire an IR-resistant phenotype (Huh7-R) (Figure 3F), which is consistent with our predicted PDK1-ALDH1 interaction probability of 0.80 or 0.99 using the random forest (RF) or support-vector machine (SVM) classifier algorithm, respectively, as shown in Figure 3D. PDK1 interacts with ALDH and directly modulate the expression and/or activity of ALDH in HCC cells. Representative western blot image and histograms of the differential expression of PDK1 and ALDH1 in adherent wild-type Mahlavu cells or their tumorsphere counterparts (Supplementary Figure S2). These data indicate, at least in part, that PDK1 directly interacts with and activates ALDH1A1, and implicates the aberrant PDK1 expression in the acquisition of IR-resistance and evasion of DNA damage by HCC cells.
Also kindly see newly included Supplementary Figure S2 and its legend, Lines 678-684.
Supplementary Figure S2. PDK1 interacts with ALDH and directly modulate the expression and/or activity of ALDH in HCC cells. (A) Representative western blot image and histograms of the differential expression of PDK1 and ALDH1 in adherent wild-type Mahlavu cells or their tumorsphere counterparts. (B) Graph showing the effect of siPDK1 on the expression level of ALDH1 mRNA in Mahlavu-R cells. (C) Representative western blot image and histograms showing the effect of siPDK1-1 and siPDK1-2 on the expression levels of PDK1 or ALDH1 protein in Mahlavu-R cells. *p<0.05, **p<0.01, ***p<0.001. Mahlavu-R, radioresistant mahlavu cells; WT, wild-type; NC, negative control.
